# Recent Advances in the Production of Genome-Edited Rats

**DOI:** 10.3390/ijms23052548

**Published:** 2022-02-25

**Authors:** Masahiro Sato, Shingo Nakamura, Emi Inada, Shuji Takabayashi

**Affiliations:** 1Department of Genome Medicine, National Center for Child Health and Development, Tokyo 157-8535, Japan; 2Division of Biomedical Engineering, National Defense Medical College Research Institute, Saitama 359-8513, Japan; snaka@ndmc.ac.jp; 3Department of Pediatric Dentistry, Graduate School of Medical and Dental Sciences, Kagoshima University, Kagoshima 890-8544, Japan; inada@dent.kagoshima-u.ac.jp; 4Laboratory Animal Facilities & Services, Preeminent Medical Photonics Education & Research Center, Hamamatsu University School of Medicine, Hamamatsu, Shizuoka 431-3192, Japan

**Keywords:** genome editing, CRISPR/Cas9, ZFNs, TALENs, rats, microinjection, electroporation, embryonic stem cell, adeno-associated virus, GONAD/*i*-GONAD

## Abstract

The rat is an important animal model for understanding gene function and developing human disease models. Knocking out a gene function in rats was difficult until recently, when a series of genome editing (GE) technologies, including zinc-finger nucleases (ZFNs), transcription activator-like effector nucleases (TALENs), and the type II bacterial clustered regularly interspaced short palindromic repeats (CRISPR)/CRISPR-associated Cas9 (CRISPR/Cas9) systems were successfully applied for gene modification (as exemplified by gene-specific knockout and knock-in) in the endogenous target genes of various organisms including rats. Owing to its simple application for gene modification and its ease of use, the CRISPR/Cas9 system is now commonly used worldwide. The most important aspect of this process is the selection of the method used to deliver GE components to rat embryos. In earlier stages, the microinjection (MI) of GE components into the cytoplasm and/or nuclei of a zygote was frequently employed. However, this method is associated with the use of an expensive manipulator system, the skills required to operate it, and the egg transfer (ET) of MI-treated embryos to recipient females for further development. In vitro electroporation (EP) of zygotes is next recognized as a simple and rapid method to introduce GE components to produce GE animals. Furthermore, in vitro transduction of rat embryos with adeno-associated viruses is potentially effective for obtaining GE rats. However, these two approaches also require ET. The use of gene-engineered embryonic stem cells or spermatogonial stem cells appears to be of interest to obtain GE rats; however, the procedure itself is difficult and laborious. Genome-editing via oviductal nucleic acids delivery (GONAD) (or improved GONAD (*i*-GONAD)) is a novel method allowing for the in situ production of GE zygotes existing within the oviductal lumen. This can be performed by the simple intraoviductal injection of GE components and subsequent in vivo EP toward the injected oviducts and does not require ET. In this review, we describe the development of various approaches for producing GE rats together with an assessment of their technical advantages and limitations, and present new GE-related technologies and current achievements using those rats in relation to human diseases.

## 1. Introduction

Rats (*Rattus norvegicus*) and mice (*Mus musculus*) belong to the same rodent family and have been the most widely used models in biomedical research for many years. However, there are several differences between these two animals. For example, the rat is larger (roughly about eight- to ten-fold) in size than the mouse, which provides a number of practical advantages, as exemplified by easier and more rapid microsurgery, multiple sampling of larger blood and tissue volumes, and precise injection of substances into blood vessels or the brain (reviewed by Kjell and Olson [1]). Additionally, mice and rats differ in their physiology, behavior, and neurology. Therefore, rats are considered as useful animals suitable for toxicological, neurobehavioral, and cardiovascular studies compared with other currently available experimental animals (reviewed by Jacob [2]).

There are some examples for the production of genetically modified (GM) rats reported approximately 20 years ago. These GM rats were transgenic (Tg) rats as a “gain of function” model to examine the role of plasma membrane calmodulin-dependent calcium ATPase isoform 4 (*PMCA4*) cDNA under control of the cardiac-specific promoter [3] or sarcoplasmic reticulum Ca^2+^-ATPase (*SERCA2a*) under control of the ubiquitous promoter [4]. Unfortunately, the production of GM rats, as exemplified by knockout (KO) or knock-in (KI) rats, has not been possible until about 10 years ago. The main reason for this is the inability to establish and maintain rat embryonic stem (ES) cells, which is much more complicated when compared with establishing and maintaining mouse ES cells. For example, in mice, ES cells were already available in the late 1980s [5,6], whereas rat ES cell production began in the late 2000s [7,8]. Since the establishment of rat ES cells, ES cell-mediated gene targeting was performed in rats for the production of KO [9,10,11,12] and KI rats [13,14,15]. For cases in which ES cells are still absent, several alternative approaches, such as transposon-mediated mutagenesis and chemical mutagenesis using N-ethyl-N-nitrosourea, have been developed to assess gene function (reviewed by Zan et al. [16]).

During the period of 2009 to 2013, genome editing (GE) technologies, such as zinc-finger nucleases (ZFNs), transcription activator-like effector nucleases (TALENs), and clustered regularly interspaced short palindromic repeat-associated protein 9 (CRISPR/Cas9) nucleases systems, all of which can induce a double-stranded break (DSB) at a specific site in the genome, appeared; since then, the production of GM rats was accelerated. Using these GE systems, many GM rats have been produced using the direct microinjection (MI) of GE components (including engineered endonucleases) into the cytoplasm or nucleus of a zygote (fertilized one-cell embryo), in vitro electroporation (EP) of isolated zygotes in the presence of GE components, and in vivo EP after the instillation of GE components into the oviductal lumen of a pregnant female (corresponding to zygote to the 2-cell stage), which is termed “genome-editing via oviductal nucleic acids delivery (GONAD) (or *improved* GONAD (*i*-GONAD))”. Furthermore, it has become possible to engineer the genome of cultured spermatogonial stem cells (SSCs), precursor cells for spermatogenesis. The transplantation of these GM SSCs into the recipient male rat testes and subsequent mating of these transplanted males with wild-type (WT) females led to the production of GM rats.

In this review, we first describe the delivery method of GE components into rat zygotes for producing GM rats. Second, we describe the novel techniques that may be potentially useful for efficient production of GM rats with multiple functions. Third, we highlight that those GE technologies are useful for producing many types of rat models for human genetic diseases.

## 2. What Is GE Technology?

As mentioned above, site-specific engineered nucleases are used in GE techniques. ZFN, TALEN, and CRISPR/Cas9 can all bind DNA and induce DSBs, which are subsequently repaired by non-homologous end joining (NHEJ) mechanisms or by homology-directed repair (HDR). In the case when the template DNA is absent, the DSB is repaired via the NHEJ pathway where insertion or deletion of nucleotides (hereafter called “indels”) frequently occurs in the cleaved area (reviewed by Harrison et al. [17] and Hsu et al. [18]). The indels often generate frameshift mutations of the amino acid sequence (thus leading to the generation of abnormal proteins) or the formation of a premature stop codon (thus leading to stoppage of protein synthesis). In contrast, in the presence of the template DNA (also called donor DNA) that is homologous to the target site, the template DNA is inserted into the cleaved area through HDR. Notably, HDR-mediated editing preferentially occurs in dividing cells. In contrast, NHEJ-mediated editing occurs in cells independent of the cell cycle [19].

ZFN consists of two components: the DNA-binding domain of ZF protein (capable of binding to the target DNA) and the DNA cleavage domain of *Fok* I endonuclease (capable of cleaving the target DNA). ZF protein contains three to six Cys2-His2 fingers, which recognize a triplet nucleotide code. After recognizing the binding site, it becomes a dimetric form, activates *Fok* I, and then cleaves DNA [20,21].

Similar to ZFN, TALEN comprises two components: transcriptional activator-like effector (TALE) protein and *Fok* I nuclease (termed TALE nucleases) [22,23,24]. TALE contains a 33–35 amino acid sequence that flanks a central DNA-binding region (amino acids 12 and 13). Notably, the design and engineering of TALENs are simpler than that of ZFN and, thus, TALENs can be engineered faster.

CRISPR/Cas9 contains two components: single-guide RNA (sgRNA) and the Cas9 protein. sgRNA comprises two components, a target-specific CRISPR RNA (crRNA) and a trans-activating crRNA (tracrRNA). These two parts can be mixed as a crRNA/tracrRNA duplex and work to guide the Cas9 protein to bind to a specific chromosomal DNA site [25,26,27]. After binding to the specific site, Cas9 cleaves one of the three bases upstream of the protospacer adjacent motif (PAM) of the DNA strand to induce a DSB. This system is different from the other GE tools such as ZFN and TALEN, and the synthesis of sgRNA is a prerequisite for this system. This development dramatically reduces both the complexity and time required for the design and implementation of gene editing.

In Table 1, a summary of GE rats produced from 2009 to 2021 is provided.

## 3. Delivery Method

For the production of GE rats, the choice of the delivery method for GE components in rat zygotes is important. The methods for the production of GE rats achieved by delivering GE reagents can be largely divided into six groups: the first is MI of GE reagents (in the form of DNA, mRNA, or protein) into the cytoplasm or nucleus of a zygote (Figure 1A); the second is in vitro EP of zygotes in the presence of GE reagents (Figure 1B); the third is in vivo EP of zygotes after instillation of GE reagents into the oviductal lumen of a pregnant female (Figure 1C); the fourth is in vitro viral infection of zygotes with recombinant adeno-associated virus (rAAV) (Figure 1D); the fifth is SSC-mediated production of GE rats (Figure 1E); the sixth is ES cell-mediated production of GE rats (Figure 1F). 

In the following sections, each of these methods is described in detail.

### 3.1. MI-Based Production of GE Rats

Since the first development of pronuclear MI in mice by Gordon et al. [154], numerous GM mice and other animals such as rats, pigs, cows, and nonhuman primates have been created (reviewed by Clark et al. [155]). These animals are mainly produced by the introduction of exogenous DNA called “transgene”, which comprises a tissue-specific or ubiquitous promoter, intron, cDNA as a gene of interest (GOI), and polyadenine tails (poly(A) site), into a male pronucleus of a zygote. The mode of transgene integration into the host chromosomes is generally random, leading to frequent transgene silencing. In contrast with this transgenic (Tg) system, GE-based modulation of endogenous target genes does not always require the chromosomal integration of the GE components.

In 2009, ZFN-based GE in rats was first reported by Guerts et al. [28], who performed a single pronuclear injection (PI) of plasmid DNA (ranging from 0.4 to 10 ng/µL) or mRNA (ranging from 0.4 to 10 ng/µL) or intracytoplasmic injection (IC) of mRNA (from 2 to 10 ng/µL) encoding ZFNs in rat zygotes (from Dahl salt-sensitive (SS), Sprague–Dawley (SD), or Fawn-hooded hypertensive (FHH) strains) to induce indels in different target genes (coding for integrated reporter green fluorescent protein (*GFP*), immunoglobulin M (*IgM*), or *Ras*-related protein Rab-38 (*Rab38*)). The PN of mRNA, PN of plasmid DNA, or IC of mRNA resulted in production of pups with the GE efficiencies of 5–29%, 6–11%, or 6–75%, respectively. These GE efficiencies shown in this first study accelerated the researchers to generate more numbers of GM rats with these technologies. For example, rat models for X-linked severe combined immunodeficiency (X-SCID) were successfully created by PI of mRNAs encoding custom-designed ZFNs (targeted to interleukin 2 receptor gamma (*Il2rg*) locus) with GE efficiencies of over 20% [29].

In 2011, TALEN-based GE in rats was first reported by Tesson et al. [32], who intended to disrupt the rat *IgM* locus to create heritable mutations that could eliminate *IgM* function. Seven out of seventy-four (10%) rat zygotes injected with plasmid DNA (0.4 to 10 ng/µL) and 51/88 (58%) of rat embryos injected with mRNA (4 and 10 ng/µL) were modified at *IgM*. Of the *IgM*-mutated mRNA-injected rats, the frequency of biallelically modified rats was dose-dependent: 8/15 (53%) of rats injected with 10 ng/µL mRNA, 5/27 (19%) at 4 ng/µL, and none at 0.8 ng/µL.

In 2011, the 2nd wave of production of ZFN-based GE was brought by Cui et al. [33], who developed a conditional KI system in rats. They used ZFN technology in embryos to introduce sequence-specific modifications (KI) through homologous recombination (HR) in SD and Long–Evans (LE) rats. Briefly, Cui et al. [33] first constructed donors with an eight-base pair (bp) *Not* I restriction site inserted between the ZFN-binding sites, flanked by ~800 bp of immediate homology on each side. Donor plasmid DNA (1 ng/µL) and respective ZFN mRNA (2.5 ng/µL) were co-injected into the pronucleus of rat zygotes. The loci targeted were rat multidrug resistance protein 1a (*Mdr1a*) and pregnane × receptor (*Pxr*) loci. When ZFN-treated fetuses were examined, 1 of 15 (7%) and 1 of 8 (13%) were GM fetuses with KI at the *Mdr1a* or *Pxr* loci, respectively. Notably, successful germ-line transmission of the indel mutations that occurred in the born founder rats was also confirmed. Next, they constructed GFP donors, replacing the *Not* I site with a 1.5-kilobase (kb) human phosphoglycerate kinase (*PGK*) promoter-driven GFP cassette. GFP is in the opposite orientation of transcription for the rat *Mdr1a* and *Pxr* loci. In the case of MI with GFP cassette into the *Mdr1a* locus of SD rats, of the 439 embryos transferred to recipients, 83 pups were born, and 2 (2%) were pups with successful KI and 21 (25%) were those having NHEJ-based mutations, suggesting that NHEJ occurs at a higher rate than targeted integration. In the eyes of founder *Mdr1a* pup No. 3, *PGK-GFP* was expressed and visually detectable.

In contrast, Brown et al. [36] first provided a system enabling the conditional ablation of a target gene by the ZFN/Cre system through MI of ZFNs into rat zygotes. Briefly, they first generated GM rats with *loxP*-flanked (floxed) alleles at the glutamate ionotropic receptor N-methyl-D-aspartate (NMDA) type subunit 1 (*Grin1*) or corticotropin releasing hormone receptor 1 (*Crhr1*) locus by co-injecting mRNA encoding two pairs of ZFNs for each gene and two donors (plasmids carrying the fragment in which *loxP* site is surrounded by left homology arm (800 bp) and right homology arm (800 bp) in both sides) for each target into the pronucleus of zygotes. Each ZFN pair cleaved in an intron flanking the exons of interest and delivered one *loxP* site. In case of *Grin1* targeting, among 80 live births, 6 (8%) were identified as those containing *loxP* in both insertion sites. Additionally, 48 animals had single *loxP* insertions, and another 17 had only NHEJ events. Brown et al. [36] also generated a GM rat expressing Cre under the control of an endogenous tyrosine hydroxylase (*Th*) promoter by inserting an internal ribosomal entry site (IRES)-Cre cassette immediately after the translational stop site of the *Th* open reading frame by pronuclear MI of a pair of ZFNs along with a donor plasmid. Of the 41 live-born pups, 9 (22%) were founders. Through crossing between these two lines (the floxed line and the Cre-expressing line), Cre-dependent and tissue-specific gene disruption successfully occurred in vivo in the bigenic offspring.

In 2013, the production of GE rats using the CRISPR/Cas9 system was reported [37,38]. This CRISPR/Cas9 system enables multigene KO in one shot of gene delivery to mice [156,157]. This property is especially beneficial for the purpose of creating disease model animals, as certain types of diseases are known to be caused by multigene defects. Li et al. [37] first attempted to generate KO rats with multiple mutations (at least in two different genes) using a CRISPR/Cas9 system. They injected two sgRNAs (12.5 ng/µL for each) (targeting rat melanocortin 3 receptor (*Mc3r*) and melanocortin 4 receptor (*Mc4r*)) and *Cas9* mRNA (25 ng/µL) into rat zygotes. The efficiency of indel formation caused by the Cas9 nuclease was considerably different because 13 of 15 (87%) F0 pups exhibited the KO phenotype for *Mc4r*, while 1 of 15 (7%) exhibited the KO phenotype for *Mc3r*. The authors concluded that a single injection can induce disruption of at least two different genes in the rat and also suggested that targeting efficiency depends on the choice of suitable gRNA. Furthermore, Li et al. [37] first demonstrated successful germ-line transmission of the KO phenotype obtained through the CRISPR/Cas9 system, although germ-line transmission has already been known using ZFNs and TALENs by others [28,32,33]. Similar to Li et al. [37], Li et al. [38] attempted to generate multiple gene mutations in rats in a germ-line competent manner using a CRISPR/Cas9 system. They designed six sgRNAs targeting six different genomic sites encoding rat Tet methylcytosine dioxygenase 1 (*Tet1*) (sgTet1-1 and sgTet1-2), Tet methylcytosine dioxygenase 2 (*Tet2*) (sgTet2-1 and sgTet2-2), and Tet methylcytosine dioxygenase 3 (*Tet3*) (sgTet3-1 and sgTet3-2) to increase the likelihood of successful targeting. Co-injection of sgRNAs for the three *Tet* genes (sgTet1-1, sgTet2-1, and sgTet3-1) together with *Cas9* mRNA into rat zygotes resulted in the generation of a total of 22 newborn pups. Of these pups, 13 (59%) contained mutations of all three *Tet* genes. Notably, ~60% of them had a biallelic or monoallelic mutation for each targeted locus. Li et al. [38] also demonstrated that mutations in the founder rats were transmitted to the next generation. 

Ménoret et al. [35] demonstrated, for the first time, that meganucleases—naturally occurring restriction enzymes derived from lower-order animals and plants—can be used as a fourth editing platform. They microinjected a plasmid encoding an engineered meganuclease for recombination, activating gene 1 (*Rag1*) into the pronuclei of rat zygotes. Of the microinjected zygotes, 0.6% were found to be mutated rats. However, this technique exhibits design complexity and associated costs, making it accessible to only a few laboratories.

Although MI of GE reagents into zygotes is one of the most useful tools for generating GM animals, it has advantages as well as disadvantages. The advantages of this technology include the delivery of known quantities of nucleic acids into a zygote irrespective of the type of zygote and the introduction of a large-size cargo (carrying a GOI), which is a significant limiting factor when using viral vectors for gene delivery, as suggested by Sato et al. [158]. The disadvantages of this technology include the requirement of a micromanipulator system and egg transfer (ET) to pseudo-pregnant females for allowing the further development of MI-treated embryos. 

Notably, the detailed protocols for MI-based GE in rats have been described by Shao et al. [159], Ménoret et al. [160], and Tesson et al. [161].

### 3.2. In Vitro EP-Based Production of GE Rats

EP is known to be a useful and powerful gene delivery tool which can enable the transfer of exogenous substances (i.e., DNA, mRNA, and protein) into a cell by forming transient pores into the cell membranes under electrical stimulation in vitro and in vivo (reviewed by Young and Dean [162]).

In 2014, Kaneko et al. [43] first demonstrated that rat zygotes can be genome-edited by in vitro EP, which is shown in Figure 1B, schematically. Since then, many researchers have successfully induced gene edits using this technology in mice [163,164,165,166,167,168], rats [95,111], bovines [169], and pigs [170,171,172]. The merit of this technology is that it is simple, rapid, and convenient for GE in zygotes, compared to the previous MI-based technique. Notably, approximately 30 to 50 zygotes can be edited with one shot of in vitro EP. Furthermore, EP only requires a square pulse generator, called an electroporator, and not a more expensive micromanipulator system. According to Kaneko et al. [43], intact F344/Stm rat zygotes were subjected to in vitro EP using an NEPA21 electroporator (NEPA GENE Co. Ltd. Chiba, Japan) in the presence of a phosphate-buffered saline (PBS)-based solution containing ZFN (40 μg/mL) mRNA (targeted to the rat *Il2rg* gene) under the EP condition of a poring pulse (Pp) (voltage: 225 V; pulse interval: 50 ms; pulse width: 1.5 and 2.5 ms; number of pulses: 4). The NEPA21 electroporator employs a 3-step pulse system. In the first step, the Pp makes micro-holes in the ZP and oolemma. In the second step, several first transfer pulses (Tp) transfer mRNA into the cytoplasm. In the third step, the polarity-changed second Tp increase the chance of transferring mRNA into embryos. When rat embryos were electroporated with a pulse width adjusted to 1.5 or 2.5 ms, 73 and 75% of the resulting offspring, respectively, had an edited *Il2rg* locus. Notably, they reported no appreciable reduction in the developmental ability of the EP-treated embryos. Furthermore, the germ-line transmission of the ZFN EP-induced editing was also confirmed in the next generation. They named this technology “Technique for Animal Knockout system by Electroporation (TAKE)”. Since then, many GE rats have been produced using EP-based GE technology [54,90,91,95,111,117,121,125,131,140,141,144]. Notably, in vitro EP can also be applied to frozen mouse and rat zygotes for the efficient introduction of CRISPR components, without affecting the embryo viability or development [173,174]. This will facilitate GE rat production in a more convenient manner.

At the initial stages of GE rat production, mRNAs encoding ZFNs, TALENs, or Cas9 have been frequently used [28,29,30,31,32,33,34,36,37,38,39,40,41,42]. However, mRNA transfection still relies on the translation of the mRNA into protein, resulting in variability of GE efficiency with timing. In the case of CRISPR/Cas9-based GE, instead of using *Cas9* mRNA, the use of the Cas9 protein allows transient and faster editing [175]. For example, the recombinant Cas9 protein and the chemically synthesized guide RNA (gRNA) can be combined on the bench before being delivered into embryos; this Cas9-gRNA complex, which is called a ribonucleoprotein (RNP), is active immediately. If the RNP enters the nuclei of cells, GE should commence quickly, which will, in turn, lead to increased GE efficiency. Using this RNP-based GE system, a number of GE rats have been produced to date [57,58,68,90,91,95,111,113,115,117,121,126,131,142,150].

Although in vitro EP enables the effective delivery of a short (<1 kb) single-stranded DNA (ssDNA) or synthetic single-stranded oligodeoxynucleotide (ssODN) into mouse zygotes, it has been difficult to introduce longer (>1 kb) double-stranded DNA (dsDNA) such as plasmid DNA. Only a few laboratories have successfully achieved this, although extensive exploration of optimal EP conditions, which is very laborious and time-consuming, is strictly required [176]. Notably, Bagheri et al. [177] overcame this limitation by first injecting all GE components, including long plasmid-sized DNA templates, into the sub-zona pellucida (ZP) space, called the “peri-vitelline space”. Therefore, they were retained, supporting subsequent in vitro EP. Bagheri et al. [177] suggest that this simple and well-tolerated method achieves intracellular reagent concentrations sufficient to provide precise gene edits.

As for the HDR-based KI by in vitro EP, Miyasaka et al. [95] first designed long single-stranded DNA (lssDNA) (constructed using a simple method including custom-made plasmids using nicking endonucleases as previously reported by Yoshimi et al. [64]) as the HDR template. The lssDNA comprises a targeted exon (exon 2 of the rat vesicle-associated membrane protein-associated protein B/C (*Vapb*) gene with a P56S mutation, which is associated with amyotrophic lateral sclerosis (ALS) in humans) flanked by two *loxP* sites. In vitro EP of rat zygotes was performed in the presence of *Cas9* mRNA (400 ng/μL), lssDNA (40 ng/μL), and two gRNAs (200 ng/μL) in a chamber with 40 μL of serum-free media (Opti-MEM) with a 5-mm gap electrode in a NEPA21 electroporator under the following conditions: Pp of voltage 225 V, pulse width—2.0 ms, pulse interval—50 ms, and number of pulses—4. The first and second Tp were of voltage 20 V, pulse width—50 ms, pulse interval 50—ms, and number of pulses—5. Consequently, of six pups born, three showed floxed alleles with the P56S mutation. They called this technology “CRISPR with lssDNA inducing conditional knockout alleles (CLICK)”. The CLICK technology will be useful as a tool for producing GM zygotes carrying transgenes driving tissue-specific Cre recombinase, which, in turn, will be applicable to the one-step generation of conditional KO animals.

When compared to the MI-based technique, in vitro EP allows the simultaneous processing of many zygotes in a short time (e.g., a batch of 30 to 50 zygotes in few seconds) without requiring expensive equipment and operators with extensive training and expertise. The disadvantages of this technology include the requirement of ET of the EP-treated embryos to pseudo-pregnant females to allow further development of the treated embryos. 

Notably, the detailed protocols for in vitro EP-based GE in rats have been reported by Kaneko [178] and Remy et al. [179].

### 3.3. GONAD-Based Production of GE Rats

In 2015, a novel method (termed “GONAD”) enabling in situ CRISPR/Cas9-based GE toward early embryos was first reported by Takahashi et al. [180] using mice. In this system, both *Cas9* mRNA (up to 1.1 μg/μL) and sgRNA (0.6 μg/μL) were co-injected into the oviductal lumen of pregnant females at Day 1.4 of pregnancy (corresponding to the 2-cell stage). Technically, this was simply achieved by inserting a glass micropipette through the oviductal wall under observation using a dissecting microscope, as shown in Figure 1C schematically. Next, a small amount of a solution (1–1.5 µL) containing GE components was injected into the oviductal lumen with the aid of a mouthpiece-controlled micropipette. Subsequently, in vivo EP was performed using tweezer-type electrodes and a square-wave pulse generator (T820; BTX Genetronics Inc. San Diego, CA, USA) under the EP condition of eight square-wave pulses with a pulse duration of 5 ms and an electric field intensity of 50 V. GE occurrence in the 2-cell embryos can be later checked by molecular analysis of the mid-gestational fetuses dissected after the GONAD. The KO efficiency of these fetuses was approximately 29%. Later, this technology was elaborated by the same group, who injected a solution containing RNP (comprising Cas9 protein (1 µg/µL Integrated DNA Technologies, Inc. Coralville, IA, USA) and sgRNA (30 µM)) (for KO experiment), RNP + ssODN (1–2 µg/µL) (for KI experiment), or RNP + ssDNA (0.85–1.4 µg/µL) (for KI experiment) at Day 0.7 of pregnancy (corresponding to the late 1-cell stage) [181]. In vivo EP using the NEPA21 electroporator was performed under the following conditions: Pp: 50 V, 5-ms pulse, 50-ms pulse interval, three pulses, 10% decay (±pulse orientation); Tp: 10 V, 50 ms pulse, 50 ms pulse interval, three pulses, and 40% decay (±pulse orientation). This modification resulted in 97% of the embryos exhibiting indels in the target locus and approximately 50% of embryos containing KI alleles. Thus, this improved technology was re-named the “*improved* GONAD (*i*-GONAD)”. Since the successful development of GONAD/*i*-GONAD in mice, the technology has proven useful for other species, such as rats [90,91,121] and hamsters [182]. 

Successful *i*-GONAD in rats was first reported in 2018 [90,91]. For examples, Takabayashi et al. [90] demonstrated that the appearance of non-pigmented eyes in fetuses (showing the KO phenotype due to indels at the tyrosinase gene (*Tyr*), a gene coding for protein essential for eye pigmentation) was induced with an efficiency of 56%, when F1 embryos from pigmented Brown Norway (BN) × albino SD rat crosses were used for *i*-GONAD targeting of *Tyr*. In this case, the in vivo EP condition used was almost the same as that used for *i*-GONAD in mice. They also tested the possibility of KI (targeted *Tyr* locus) using albino Lewis (LEW) rats. The *i*-GONAD-mediated KI efficiency, which was evaluated by the presence of fetuses with pigmented eyes, was as low as ~5%. Kobayashi et al. [91] provided data similar to that of Takabayashi et al. [90] using the NEPA21 electroporator under the following EP conditions: Pp: 30, 40, or 50 V, 5-ms pulse, 50-ms pulse interval, number of pulses 3, and 10% decay (±pulse orientation); Tp: 10 V, 50-ms pulse, 50-ms pulse, number of pulses 6, and 40% decay (±pulse orientation). According to Kobayashi et al. [91], intraoviductal injection of RNP (1 μg/μL of Cas9 protein +30 μM of sgRNA) targeted to *Tyr* into oviducts of pregnant albino WKY females (which had been successfully mated with dark agouti (DA) males) or pigmented DA females (which had been successfully mated to Wistar–Kyoto (WKY) males) resulted in the production of GE pups with efficiencies of 59% or 42%, respectively, when 50 V of the voltage (Pp) was employed. These results suggest no significant difference in the gene-editing efficiency between these two strains. They also attempted to recover the coat-color mutation in WKY females using an ssODN-based KI approach. Consequently, the KI efficiency was 27% in the pups born. They named this rat-based *i*-GONAD as “rGONAD”.

For performing *i*-GONAD (or rGONAD) in rats, the exploration of optimal EP conditions is strictly required. For example, when a current of >500 mA was employed, albino SD and albino LEW rats were successfully genome-edited; however, the acquisition of offspring derived from pigmented BN rats (fetuses/newborns) failed. In contrast, *i*-GONAD under a current of 100–300 mA using the NEPA21 electroporator led to the production of GE BN rats with efficiencies of 75% to 100% [90]. Similarly, *i*-GONAD under a current of 150–200 mA using another electroporator CUY21EDIT II (BEX Co., Tokyo, Japan) led to the production of genome-edited BN rats with efficiencies of 24% to 55% [121].

The advantages of this technology are that GONAD/*i*-GONAD requires an electroporator but no other special equipment. Furthermore, no recipients are required unlike the case of MI- or in vitro EP-based GE rat production. Disadvantages of this technology are that technical skill may be required for intraoviductal injection of a solution into the lumen of oviducts under a dissecting microscope using a mouthpiece-controlled glass micropipette. 

The detailed protocols for GONAD/*i*-GONAD have been reported by Gurumurthy et al. [183] and Ohtsuka and Sato [184] in mice. The GONAD/*i*-GONAD-based production of GE rats is also possible using the same approach shown in mice. Notably, Namba et al. [185] demonstrated the protocols for GONAD/*i*-GONAD in rats.

### 3.4. rAAV-Mediated Production of GE Rats

rAAVs are non-pathogenic and non-enveloped ssDNA viruses enabling highly efficient transduction of both dividing and non-dividing cells (reviewed by Daya and Berns [186]). Unlike other viral vectors such as retroviruses and adenoviruses, rAAVs have been used for manipulating the target genome sequences through simple co-incubation with ZP-enclosed preimplantation mammalian embryos for the production of GM animals such as mice [168,187,188,189,190,191]. For example, Mizuno et al. [187] observed that, among the rAAVs tested, the rAAV with serotype 6 (which is hereinafter referred to as rAAV-6) showed the highest transduction efficiency when ZP-intact murine embryos were co-cultured with rAAVs. More importantly, Mizuno et al. [187] demonstrated successful CRISPR-based KI in mice. As shown in Figure 1D, zygotes were first subjected to in vitro EP in the presence of RNP and then infected with rAAV-6 carrying a 1.8-kb GFP expression cassette flanked by two 100-bp *Rosa26* homology arms. The KI efficiencies in the *Rosa26* locus were 16% and 6%, when assessed at the blastocyst and newborn stages, respectively. Notably, Mizuno et al. [187] demonstrated that both ZP-intact rat and bovine embryos were effectively infected by rAAV-6, suggesting that rat genome can be effectively edited through the simple incubation of ZP-intact rat zygotes with a solution containing rAAVs.

This rAAV-based gene delivery system to early embryos was also observed in vivo. Yoon et al. [188] injected a solution containing two vectors, rAAV6-*Cas9* (carrying *spCas9* gene derived from *Streptococcus pyogenes*) and rAAV6-g*Tyr* (carrying gRNA expression unit targeted *Tyr*), into the oviducts of pregnant female mice at Day 0.5 of pregnancy, similar to GONAD/*i*-GONAD. Out of the 29 pups that were obtained, three (10%) were found to have indels. All mutated founder mice generated albino offspring, indicating germ-line transmission. These results suggest that AAV is a powerful tool for inducing GE in the ZP-enclosed early embryos in vivo. According to Sato et al. [158], this in vivo approach is referred to as “AAV-based GONAD”.

The advantages of this technology are that no special equipment is required in the case of using the AAV-based GONAD. Notably, the use of highly concentrated rAAV would be one of the key factors for achieving a high rate of transduction of preimplantation embryos. According to Yoon et al. [188], zygotes treated with 6 × 10^8^ genome copies (GCs) for one day in vitro resulted in live births with 100% indel frequency after ET. However, the editing frequency dropped to 25% for fetuses and 20% for newborns at 6 × 10^7^ GCs. No edited animals were detected from the 6 × 10^6^ GC treatment group. These results clearly suggest that gene delivery and GE efficiency are rAAV dose-dependent. 

### 3.5. SSC-Mediated Production of GE Rats

Cultured SSCs can display long-term sperm-forming potential when they are transplanted into the seminiferous tubules (STs) of sterile rat testes (shown in Figure 1E). It is now possible to culture SSCs for a long time and perform gene modification toward SSCs derived from mice and rats (reviewed by Sato et al. [158]). 

Chapman et al. [60] first demonstrated that CRISPR/Cas9-mediated GE can be successfully applied in SSCs to create KO rats. They used SSCs isolated from tg*GCS-EGFP* rats (with the genetic background of SD) showing germ cell-specific expression of enhanced green fluorescent protein (EGFP) [192]. The SSCs were then transfected with the pX330 vector (conferring expression of both Cas9 and gRNA targeted epithelial stromal interaction 1 (*Epsti1*) gene) together with a neomycin resistance gene expression plasmid. The surviving clones after G418 selection were then transplanted into the STs of recipient sterile rats to promote spermatogenesis. When the recipient rats transplanted with G418-selected cells were mated with WT females at ~65 days post-transplantation, a total of 87 pups were obtained. Of these, ~10% (9) were found to harbor *Epsti1* mutants. A similar attempt was also made by Noto et al. [102], who successfully obtained immunodeficient rats by grafting recombination activating gene 2 (*Rag2*) KO SSCs into rat testes. 

### 3.6. ES Cell-Mediated Production of GE Rats

As mentioned previously, acquisition of a rat ES cell line is now possible. Yamamoto et al. [14] first succeeded in producing KI rats using CRISPR/Cas9-engineered ES cells. Initially, they obtained GM ES cell clones using a conventional gene-targeting technology but failed because of no recombinant ES cells were obtained. Next, in vitro EP of a Cas9-expressing plasmid, gRNA-expressing plasmid, and KI vector (carrying two homology arms) in rat ES cells was performed. Consequently, the number of drug-resistant colonies increased and the HR efficiency was considerably enhanced from 0 to 36%. These recombinant clones resulted in the successful production of chimeras, which were later proven to have the ability to transmit the KI allele to their offspring after mating with WT rats (as shown in Figure 1F). According to Yamamoto et al. [14], the methodology presented can overcome the problem encountered in conventional gene targeting that prevented the production of humanized rats. Later, successful GM rat production through ES cells carrying CRISPR/Cas9-based KI of the *Discosoma* coral red fluorescent protein (*dsRed*) gene at the Sry-box transcription factor 10 (*Sox10*) locus was reported by Chen et al. [15]. 

## 4. Other Techniques and Factors to Modify the Rat Genome

As shown above, GE tools such as ZFNs, TALENs, and CRISPR/Cas9 are considered useful for the production of GE rats. However, several techniques and factors still influence their performance, which must be addressed. These include the allele-specific GE using ssODNs, large genomic fragment deletion, ssODN-mediated KI without the need to construct homology arms, the use of inhibitors for efficient KI, the importance of choosing the appropriate gRNA for efficient KI, and base editor systems enabling precise genome modifications. 

In this section, these techniques or factors are described in detail.

### 4.1. Allele-Specific GE for the Correction of Mutated Phenotypes

Numerous reports have demonstrated that CRISPR/Cas9 editors combined with ~100 bp ssODNs, instead of using a relatively long template (~1 kb), allow for precise HDR-mediated GE in mice [156,157,193]. These results suggest that ssODNs co-injected as donor templates preferentially support the activation of HDR relative to the NHEJ pathway.

To explore the applications of ssODN-mediated GE in rats, Yoshimi et al. [46] successfully used ssODNs to manipulate the coat-color genetics of F344 rats. They used a WT ssODN to exchange a single base pair in *Tyr*, thus reverting the albino allele back to the WT sequence and restoring a non-albino pigmentation pattern. A similar strategy using ssODN for the correction of mutated phenotypes in rats was also employed by others [64,90,91,106,114,125,126,127,129,141,142].

### 4.2. Large Genomic Fragment Deletion

The GE system has enabled the creation of targeted deletion spanning the large genomic region over 10 kb in size via the direct MI of two sgRNAs (~10–30 ng/μL) and *TALEN* mRNA (45 ng/μL) or *Cas9* mRNA (~67–100 ng/µL) into the pronuclei or cytoplasm of mouse zygotes [194,195]. Wang et al. [58] achieved 53-kb deletion of a fragment spanning long non-coding RNA (lncRNA) GM14005 genomic DNA by injecting the Cas9 protein (30 nM) into the cytoplasm of mouse zygotes together with two sgRNAs (12.5 ng/µL). Among five F0 pups, one founder was confirmed as a mouse with a large deletion. Wang et al. [58] also showed that the 53-kb deletion was efficiently transmitted to the F1 generation.

Yoshimi et al. [46] succeeded in the deletion of a 7-kb fragment from the rat genome. They used two flanking gRNAs to delete a 7098-bp endogenous retroviral element (ERV) within the first intron of the *Kit* gene, using ssODN containing homology arms on either side of the junction to bridge across the deletion (Figure 2). They injected 100 ng/μL *Cas9* mRNA, 50 ng/μL gRNAs (targeted *Kit*), and 120 bp ssODNs (50 ng/μL) into the pronuclei of Long–Evans hooded (LEH) rat zygotes. Thus, GE rats exhibited restoration of the function of the *Kit* gene with a non-hooded pigmentation pattern. The frequency of the appropriately edited pups was 4%, showing the feasibility of performing template-directed correction of mutated portions using ssODNs. 

If both DSBs occur at the same time, the result will be a large deletion of the region of interest. To the best of our knowledge, the biggest deletion achieved to date in rats is 24.4 mega base pairs (Mb) [196].

### 4.3. ssODN-Mediated KI with CRISPR/Cas9 for Large Genomic Regions in Rat Zygotes

Although the CRISPR/Cas9 system is recognized as a powerful tool for generating GM animals, the creation of targeted KI animals via HR is still challenging. Yoshimi et al. [64] developed a novel KI method without the need to construct homology arms, thus simplifying the process of genome engineering in living organisms. In other words, they tried to knock-in longer plasmids (including the 1.2-kb chicken β-actin-based promoter (called “CAG”) and a 0.8-kb GFP cassette sequence) at the targeted site (rat *Rosa26* locus) (Figure 3). In more detail, they constructed two gRNAs: one targeting the rat *Rosa26* locus to cleave genomic DNA, and the other targeting the 5′ end of the CAG promoter sequence for concurrent cleavage of the plasmid DNA (Figure 3). Two 80-bp ssODNs (ssODN-1 and -2) were designed to ligate the two cut ends, as shown in (Figure 3). A mixture of 100 ng/μL of the *Cas9-poly(A)* mRNA (which appended an 81-bp poly(A) to the transcription terminator to increase the efficacy of GE with the CRISPR/Cas9 system), 50 ng/μL of each of the two gRNAs (gRNA:rRosa26 and gRNA:CAGGS), 50 ng/μL of each of the two ssODNs (ssODN-1 and -2), and 5 ng/μL of the CAG-GFP plasmid was co-microinjected into the pronuclei of rat zygotes. Of the 17 pups obtained, four (#6, #7, #8, and #11) were selected as candidates with successful KI. Sequence analysis revealed that #11 had accurate conjunction at both of the two cut ends, #6 had a 6-bp deletion on one side, and #7 had several base-pair deletions and insertions on both sides. They confirmed germ-line transmission of the GFP allele. No off-target effect was observed at any candidate sites in these rats. Furthermore, uniform expression of the GFP gene in most cells and tissues examined was confirmed. Based on these observations, this strategy was called “two-hit by gRNA and two-oligo with plasmid (2H2OP)”.

### 4.4. Use of Inhibitor for Efficient KI

KI is an event of precise modifications at a target locus and is often mediated by HR in the presence of donor DNA, as exemplified by ssODN. Generally, HDR-mediated KI is more difficult than NHEJ-based indels. For example, in proliferating human cells, NHEJ has been reported to repair 75% DSBs, whereas HDR repaired the remaining 25% [197]. To enhance the HDR efficiency, several approaches are now being attempted. In the case of a CRISPR/Cas9-mediated KI event, Scr7, which was first identified as an anti-cancer compound and was considered a potential NHEJ inhibitor, was used to increase the HR-mediated precise genetic modification [198,199]. For example, co-injection of murine zygotes with a mixture containing *Cas9* mRNA, sgRNA, template ssODNs, and Scr7 significantly improved the efficiency of HDR-mediated insertional mutagenesis [198]. Chu et al. [200] also demonstrated the usefulness of Scr7 for abolishing NHEJ activity and increasing HDR in both human and mouse cell lines. However, the function of Scr7 in promoting HDR remains controversial. Some researchers have demonstrated that Scr7 failed to increase HDR rates in rabbit embryos [201] and porcine fetal fibroblasts [202]. 

In rats, Ma et al. [68] first demonstrated that both Scr7 and Cas9 protein can increase the precise modification in rat embryos when the fatty acid binding protein 2 (*Fabp2*) and dysbindin (dystrobrevin binding protein 1) domain containing 1 (*Dbndd1*) loci were selected to perform KI of a fragment containing Cre or Cre-ERT2 (Cre recombinase-estrogen receptor T2) gene, respectively. For example, when a mixture containing *Cas9* mRNA (25 ng/µL), sgRNA (10 ng/µL; targeting *Fabp2*), plasmid template (4 ng/µL; for *Fabp2*), and Scr7 (0, 0.5, 1 or 2 μM) was injected into both the cytoplasm and pronuclei of rat zygotes, two (18%) of 11 pups born exhibited KI in the absence of Scr7. In contrast, three (33%) of 9 and 11 (39%) of 28 pups were found to exhibit KI after MI with 2 or 1 μM Scr7, respectively.

However, the opposite results were recently achieved by Takabayashi et al. [90] and Aoshima et al. [141]. Takabayashi et al. [90] performed *i*-GONAD using a solution containing 1 μM Scr7, RNP (1 μg/μL of Cas9 protein +30 µM of sgRNA), and 2 µg/µL of ssODN resulted in the production of 27 fetuses, of which only two (7%) had pigmented eyes. Notably, in the absence of Scr7, *i*-GONAD-mediated KI efficiency was as low as ~5%, suggesting that addition of Scr7 appeared not to affect *i*-GONAD-mediated KI efficiency in rats. Aoshima et al. [141] examined whether commercially available KI-enhancing drugs (including Scr7, L755,507, RS-1, and HDR enhancer), all of which are known to enhance KI efficiency, can increase the KI efficiency using *i*-GONAD in rats. Aoshima et al. [141] demonstrated that some drugs (e.g., Scr7, L755,507, and HDR enhancer) were found to be slightly effective in vivo, but their effects were not statistically significant when compared to those in the control group (without any KI-enhancing drugs). 

### 4.5. Choice of gRNA Is Very Important for Achieving High KI Efficiency in Rats

According to Raveux et al. [203], ssODN-mediated KI efficiency varied from 0% to 40%, depending on the type of Cas9, the injection site, the length of the homology arm, and the gRNA-binding site. The nucleotide composition of a target sequence is one of the most important factors determining KI and KO efficiency. For example, Graf et al. [204] screened the published gRNA activity datasets and demonstrated that TT- and GCC-motifs located in the four PAM-proximal bases of the targeting sequence (the efficiency-modulating sequence) were sufficient to block CRISPR/Cas9-mediated KI. Moreover, the presence of these motifs in gRNAs resulted in a 10-fold reduction in gene KO frequencies.

Aoshima et al. [141] constructed three gRNAs showing high efficiencies, ranging from 51% to 57%, based on the CHOPCHOP predictions [205]. These three gRNAs (termed crRNA1, crRNA2, and crRNA3) used here recognize the *Tyr* sequence containing a point mutation (G to A) and overlap with each other (Figure 4). When KI-based *i*-GONAD was performed using ssODN (2 μg/μL), Cas9 protein (1 μg/μL), and gRNA (crRNA1, 2 or 3) (30 μM), the KI efficiency in fetuses generated after *i*-GONAD with crRNA2 was significantly higher (24%) than that obtained after *i*-GONAD with crRNA1 (5%) or crRNA3 (0%). The discrepancy between predicted and actual efficiencies was unexpected. Notably, no inhibitory sequences exist in the gRNAs used, unlike the suggestion of Graf et al. [204]. Furthermore, in the case of in vitro GE, the GE efficiency is known to vary, with predicted efficiencies typically higher than observed levels [156,206,207]. The data of Aoshima et al. [141] suggest the importance of selecting the appropriate gRNA for effective KI in rats.

### 4.6. Base Editor Systems as An Efficient Tool Enabling Precise Genome Modifications

Single-base editing systems without the induction of DSB or template DNA, mediated by cytidine deaminase combined with Cas9 nickase (nCas9) or dead Cas9 (dCas9), were recently developed to modify mammalian genomic DNA. This system is called the “cytosine base editor (CBE)” which achieves C·G to T·A conversion with a window of approximately five nucleotides [208,209]. Furthermore, Gaudelli et al. [210] developed a new base-editing (BE)-based system, called the “adenine base editor (ABE)”, which efficiently enables the induction of A·T to G·C conversion through fusion of an engineered tRNA adenosine deaminase (ecTadA) with nCas9. Thus, together with CBE, ABE enables the introduction of all four nucleotide transitions (C to T, A to G, T to C, and G to A) in the target genomic sequence [210]. Indeed, using these systems, the correction of genetic disease has been successfully performed in mice [211,212].

With respect to the production of GM rats using the BE systems, Yang et al. [97] reported the successful production of KO rats through MI of ABE mRNA together with sgRNA (targeted glycogen storage disease type 2 (*Gsd2*) gene) to generate an animal model for glycogen storage disease type II. Consequently, 85% (28/33) of rats had single or multiple A > G substitutions between position A3 and A7 in the target protein (GSD2), leading to I646V or D645G mutations. Similarly, Ma et al. [104] injected ABE mRNA together with the sgRNA-targeted hemogen (*Hemgn*), N-deacetylase (*Ndst1*), or N-sulfotransferase 4 (*Ndst4*) gene into rat zygotes. For *Hemgn* targeting, the zygotes were microinjected with ABE mRNA and sgRNA (targeted *Hemgn*); a total of 99 injected zygotes were transferred to three pseudo-pregnant female rats, and 15 pups were born. Consequently, fourteen rats (14/15) contained an A to G conversion at the 14th base distal from the PAM, indicating the high base-editing efficiency. According to Ma et al. [104], these BE tools can provide a much safer approach compared with the WT CRISPR/Cas9 system for the gene correction of human disease.

Recently, Qi et al. [153] demonstrated that a bacterial toxin deaminase (DddA) from *Burkholderia cenocepacia*, which can convert cytosine to uracil specifically within dsDNA, can be used for mediating mitochondrial DNA (mtDNA) editing in rats. It has been demonstrated that the toxin domain of DddA (DddAtox, 1264–1427 amino acids) is engineered and incorporated with the mitoTALE system to efficiently achieve C ∙ G-to-T ∙ A conversion in mtDNA of human cell lines [213]. This system is therefore termed “DddA-derived cytosine base editor (DdCBE)” [213]. Qi et al. [153] injected mRNAs of a DdCBE pair with the best performance in rat zygotes, but the DdCBE-mediated BE failed. They reasoned that this failure was due to the inactive mtDNA replication in zygotes because this system works during the period when mtDNA replication is active. To overcome this problem, the DdCBE pair was cloned into the *piggy*Back (PB) transposon vector, and then, the resulting PB vector was co-injected into rat zygotes with PB transposase mRNA. Thus, one out of four pups obtained was successfully edited with 36% efficiency for one target locus. According to Qi et al. [153], almost all DdCBE pairs failed to integrate into the rat genome by the transposon system; however, the expression of the DdCBE pair from the PB vector might have lasted longer than that from mRNA injection, facilitating C ∙ G-to-T ∙ A conversion in mtDNA. Using the DdCBE system, animals with precise mtDNA mutations can be produced on demand.

## 5. Disease Models in Rats

For a decade, the advent of genetic engineering tools such as ZFNs, TALENs, and CRISPR/Cas9 has led to a revolution in obtaining specific and targeted genetic mutations in rats for the study of human genetic diseases. Several important genetic diseases have been modeled in rats. A brief description of the most useful models is provided below.

### 5.1. Models for Cardiovascular Diseases

Monoallelic mutations in the gene encoding bone morphogenetic protein receptor 2 (*Bmpr2*) are the main genetic risk factors for heritable pulmonary arterial hypertension (PAH) with incomplete penetrance. Several *Bmpr2* Tg mice have been reported to develop mild spontaneous PAH. Ranchoux et al. [53] generated *Bmpr2* KO rats using the ZFN technology. The resulting KO rats with a heterozygous 140 bp deletion in the first exon of *Bmpr2* displayed intense pulmonary vascular remodeling. The same group [116] also generated rat lines with mutations (deletion of 71 bp in exon 1 of *Bmpr2*) and showed that the heterozygous rats developed age-dependent spontaneous PAH with a low penetrance (16–27%), similar to that in humans. They concluded that this new genetic rat model represents a promising tool to study PAH pathogenesis. Concerning the pathogenesis of PAH, mutations in the potassium channel subfamily K member 3 (*KCNK3*) gene, which encodes an outward rectifier K+ channel, have been identified in PAH patients. Lambert et al. [109] generated *Kcnk3* KO rats through the CRISPR/Cas9 technology. The resulting rats developed age-dependent PAH associated with low serum-albumin concentrations. Lambert et al. [109] concluded that KCNK3 loss of function is a key event in PAH pathogenesis.

### 5.2. Models for Neurological Diseases

Animal models are required to understand the pathogenesis of autism spectrum disorder (ASD). Despite the apparent advantages of mice for neural studies, rats have not been widely used for ASD studies, probably owing to the lack of convenient genome manipulation tools. Hamilton et al. [42] generated two rat models for ASD, one syndromic and one non-syndromic model, through the ZFN system, by destroying a gene (*Fmr1*) coding the Fragile X mental retardation protein (FMRP), the protein responsible for the pathogenesis of Fragile X syndrome (FXS), or a gene (*Ngln3*) coding for Neuroligin3 (NLGN3), a member of the neuroligin synaptic cell-adhesion protein family, respectively. Both FMRP and NLGN3 have been implicated in the pathogenesis of human ASD. Both KO rat lines exhibited abnormalities in ASD-relevant phenotypes including juvenile play, perseverative behaviors, and sensorimotor gating, suggesting the utility of these rats as genetic models for investigating ASD-relevant genes. Later, Tian et al. [77] produced *Fmr1* KO rats through the CRISPR/Cas9 technology targeted to exon 4 of *Fmr1*. Consistent with the previous reports, deletion of the *Fmr1* gene in rats specifically impairs long-term synaptic plasticity and hippocampus-dependent learning in a manner resembling the key symptoms of FXS.

DEP-domain containing 5 gene (*Depdc5*), encoding a repressor of the mechanistic target of rapamycin complex 1 (mTORC1) signaling pathway, has recently emerged as a major gene mutated in familial focal epilepsies and focal cortical dysplasia. Marsan et al. [67] produced a *Depdc5* KO rat using the TALEN technology. Homozygous KO embryos died from Day 14.5 of pregnancy. Heterozygous KO rats developed normally and exhibited no spontaneous electroclinical seizures; however, they showed altered cortical neuron excitability and firing patterns. This rat model is considered a relevant model to study pathogenic mechanisms underlying those disorders.

One subtype of ASD is associated with mutations in the methyl-CpG-binding protein 2 (*Mecp2*) gene, causing an X-linked neurodevelopmental disorder called Rett syndrome (RS). Patients with RS have cognitive defects and circadian clock dysfunction, as exemplified by abnormal sleep patterns. According to Zhai et al. [61], there is an urgent need for new animal models for RS because the existing *Mecp2* KO mouse models fail to fully mimic the pathogenesis and symptoms of patients with RS. *Mecp2* KO rats were successfully produced by the two groups using ZFN [62] and CRISPR/Cas9 [61] technologies. The resulting KO rats exhibited significant abnormalities in growth (body weight loss) as well as behavioral function (anxiety tendency and cognitive deficits) [61,62]. Because these phenotypes well recapitulate the major symptoms of RS patients, these *Mecp2* KO rats will provide an alternative tool for future studies of MeCP2 functions.

Mutations in fused in sarcoma (*Fus*), a nuclear DNA/RNA-binding protein, cause familial ALS and occasionally frontotemporal dementia. Zhang et al. [87] produced KI rats expressing a *Fus* point mutation (R521C) as a model for ALS using the CRISPR/Cas9 technology. The mutant animals developed adult-onset learning and memory behavioral deficits, with reduced spine density in the hippocampal neurons. Remarkably, the sleep–wake cycle and circadian abnormalities preceded the onset of cognitive deficits. These results suggest a new role of *Fus* in sleep and circadian regulation and demonstrate that functional changes in FUS could cause sleep–wake and circadian disturbances as early symptoms.

Emmert et al. [113] produced a novel rat model of X-linked hydrocephalus (XLH) by CRISPR-mediated mutation in the L1 cell-adhesion molecule (*L1cam*) gene on the X chromosome. Hemizygous male mutants developed hydrocephalus and delayed development. The mutant rats did not show reactive gliosis, but exhibited hypomyelination and increased extracellular fluid in the corpus callosum.

GABAergic dysfunctions have been implicated in the pathogenesis of schizophrenia, especially the associated cognitive impairments. The level of the GABA synthetic enzyme glutamate decarboxylase 67 kDa isoform (GAD67) encoded by the *GAD1* gene is downregulated in the brains of schizophrenia patients. Furthermore, a schizophrenia patient harboring a homozygous mutation of *GAD1* has recently been discovered. *Gad1* KO mice exhibited perinatal lethality [214], which precluded characterization at adult stages. Fujihara et al. [125] generated *Gad1* KO rats using the CRISPR/Cas9 technology. Surprisingly, 33% *Gad1* KO rats survived to adulthood, which made further characterization possible. The *Gad1* KO rats exhibited impairments in both spatial reference and working memory without affecting adult neurogenesis in the hippocampus. In addition, *Gad1* KO rats showed a wide range of behavioral alterations, such as enhanced sensitivity to an NMDA receptor antagonist, hypoactivity in a novel environment, and decreased preference for social novelty. These results suggest that *Gad1* KO rats could be a novel model for studying cognitive deficits. Furthermore, Fujihara et al. [125] claimed the necessity to check species differences in the mode of phenotype manifestation when animal models of human diseases are considered.

Angelman syndrome (AS) is a rare genetic disorder characterized by severe intellectual disability, seizures, lack of speech, and ataxia. The gene responsible for AS is the ubiquitin protein ligase E3A (*Ube3a*) gene, which encodes for ubiquitin ligase E6-associated protein (E6AP). Dodge et al. [122] generated *Ube3a* KO rats using the CRISPR/Cas9 system. The resulting KO rats phenotypically mirrored human AS with deficits in motor coordination as well as learning and memory. This model can, thus, offer a new avenue for the study of AS. 

Koster et al. [142] produced a pigmented KO rat model for lecithin retinol acyltransferase (LRAT) using the CRISPR/Cas9 system. The introduced mutation (c.12delA) is based on a patient group harboring a homozygous frameshift mutation in the *Lrat* gene (c.12delC), causing a dysfunctional visual (retinoid) cycle. The resulting KO rats exhibited progressively reduced electroretinography potentials from two weeks of age onwards and overall retinal thinning. Vision-based behavioral assays confirmed the reduced vision. These KO rats are a novel animal model for retinal dystrophy, especially for early-onset retinal dystrophies.

### 5.3. Models for Muscular Diseases

Duchenne muscular dystrophy (DMD) is an X-linked lethal muscle disorder caused by mutations in the Duchenne muscular dystrophy (*Dmd*) gene encoding dystrophin. DMD model animals, such as *Mdx* (X-linked muscular dystrophy) mice and canine X-linked muscular dystrophy dogs, have been widely utilized in the development of a treatment for DMD. However, according to Larcher et al. [51], large animal models such as dogs are expensive and difficult to handle. In contrast, *Mdx* mice only partially mimic the human disease, with limited chronic muscular lesions and muscle weakness. Their small size also imposes limitations on analyses. In this context, a rat model could represent a useful alternative because rats are bigger than mice and could better reflect the lesions and functional abnormalities observed in DMD patients.

*Dmd* KO rats were successfully produced by two groups using CRISPR/Cas9 technology targeting two exons of the rat *Dmd* [50] or using TALENs targeting exon 23 [51]. These resulting KO rats exhibited a decline in muscle strength and the emergence of degenerative/regenerative phenotypes in the skeletal muscle, heart, and diaphragm. Furthermore, these phenotypes were transmitted to the next generation [50]. Notably, *Dmd* KO rats, but not mice, present cardiovascular alterations close to those observed in humans, which are the main cause of death of patients [51].

Desminopathy is a clinically heterogeneous muscle disease caused by over 60 different mutations in the desmin (*DES*) gene. The most common mutation with a clinical phenotype in humans is an exchange of arginine to proline at position 350 of desmin leading to p.R350P. Langer et al. [126] first produced a KI rat model for a muscle disease through the CRISPR/Cas9 technology using ssODN carrying the missense mutation *Des* c.1045-1046 (AGG > CCG) in exon 6 of *Des*. While muscle weights did not differ between the mutant rats and WT rats, the levels of many muscle-related proteins such as dystrophin, syntrophin, dysferlin, and annexin A2 increased in the mutant rats, showing the phenotype of desminopathy. This rat model will be a useful tool for furthering our understanding of the disease and testing therapeutic approaches to delay disease progression.

### 5.4. Models for Pulmonary Diseases

Cystic fibrosis (CF) is characterized by airway and digestive pathology with a reduced life expectancy and is one of the most common genetic diseases in western populations. The most common mutation is the missense mutation p.Phe508del (or F508del) in the cystic fibrosis transmembrane conductance regulator (*CFTR*) gene, which leads to abnormal CFTR function and mucus accumulation. Tuggle et al. [52] first generated *Cftr* KO rats using the ZFN technology. The resulting KO rats lacked CFTR activity and exhibited abnormalities in the ileum and increased intracellular mucus in the proximal nasal septa. Airway surface liquid and periciliary liquid depths were reduced, and the submucosal gland size was abnormal. Similar results have also been observed in another *Cftr* KO rat developed by Dreano et al. [106] using the CRISPR/Cas9 technology. Furthermore, Dreano et al. [106] produced another KI rats (carrying F508del mutation). These rats showed residual CFTR activity and milder phenotype than the *Cftr* KO rats. These findings suggest that these rat models can be a CF animal model that recapitulates various aspects of the human disease. Notably, a humanized CF rat strain expressing the G551D variant was produced through KI of the human *CFTR* cDNA carrying the G551D mutation downstream of the endogenous *Cftr* promoter using the ZFN technology [130].

### 5.5. Models for Metabolic Diseases

The low-density lipoprotein receptor (*LDLR*) and apolipoprotein E (*APOE*) genes control normal levels of cholesterol and other forms of fat in the blood. A deficiency in *ApoE* is involved in several age-related fatty acid diseases. Wei et al. [56] established *ApoE* KO rats through TALEN-mediated gene targeting. After being fed with a high-cholesterol diet (HCD) for 12 weeks, the *ApoE* KO rats displayed typical dyslipidemia, although there was no obvious atherosclerotic lesion in the en face aortas. Notably, partial ligation caused the formation of plaques consisting of lipids and macrophages in carotid arteries from *ApoE* KO rats. Wei et al. [56] concluded that the *ApoE* KO rats can be a novel model for dyslipidemia. Lee et al. [110] produced *ApoE* KO rats using Cas12a (previously named Cpf1), an RNA-guided endonuclease, as a part of the CRISPR system. The resulting KO rats displayed hyperlipidemia and aortic lesions.

A deficiency in *LDLR* is a cause of familial hypercholesterolemia (FH). Zhao et al. [92] and Lee et al. [110] created *Ldlr* KO rats. These *ApoE* and *Ldlr* KO rats mimic pathological changes observed in hyperlipidemia and atherosclerosis in humans with genetic deficiencies and in normal individuals, suggesting usefulness in the research of atherosclerosis.

Melanocortin-3 and -4 receptors (MC3R and MC4R) regulate energy homeostasis. You et al. [70] generated *Mc3r* and *Mc4r* single- and double-KO (DKO) rats using the CRISPR/Cas9 system. *Mc3r* KO rats displayed hypophagia and decreased body weight, whereas *Mc4r* KO and DKO rats exhibited hyperphagia and increased body weight. All three mutants showed increased white adipose tissue mass and adipocyte size. These mutant rats will be important in defining the complicated signaling pathways of MC3R and MC4R. According to You et al. [70], both *Mc4r* KO and DKO rats are good models for obesity and diabetes research.

Hereditary tyrosinemia type I (HT1) is caused by a deficiency in fumarylacetoacetate hydrolase (FAH) enzyme. *Fah*-deficient mice and pigs are phenotypically analogous to human HT1, but do not recapitulate all chronic features of the human disorder, especially liver fibrosis and cirrhosis. Zhang et al. [63] produced *Fah* KO rats through MI of CRISPR/Cas9 components to obtain HT1 models. The *Fah* KO rats faithfully represented hypertyrosinemia, liver failure, and renal tubular damage. More importantly, they developed remarkable liver fibrosis and cirrhosis, which have not been observed in *Fah* mutant mice or pigs. These data suggest that *Fah* KO rats may be used as an animal model of HT1 with liver cirrhosis.

Pseudoxanthoma elasticum (PXE)—a heritable ectopic mineralization disorder—is caused by mutations in the ATP-binding cassette subtype C number 6 (*ABCC6*) gene primarily expressed in the liver and kidneys. These mutations result in generalized arterial calcification throughout the body in infancy. Li et al. [75] generated *Abcc6* KO rats as models of PXE using the ZFN technology. The plasma inorganic pyrophosphate (PPi) level was reduced (<30%), leading to a lowered PPi/inorganic phosphate plasma ratio. When in situ liver and kidney perfusions were performed, the PPi levels in the perfusates were significantly reduced, but those in the liver of WT rats remained high. Li et al. [75] speculate that hepatic ABCC6 may play a critical role in contributing to plasma PPi levels, identifying the liver as a target of molecular correction to counteract ectopic mineralization in PXE.

Wolfram syndrome (WS) is a rare autosomal-recessive disorder caused by mutations in the Wolfram syndrome 1 (*WFS1*) gene and characterized by juvenile-onset diabetes, optic atrophy, hearing loss, and a number of other complications. According to Plaas et al. [76], no mutant *Wfs1* mice displayed fasting hyperglycemia. In other words, previous mouse models of WS involved only partial diabetes and other disease symptoms. Plaas et al. [76] generated *Wfs1* KO rats, in which exon 5 of the *Wfs1* gene was deleted, resulting in a loss of 27 amino acids from the WFS1 protein. The resulting KO rats showed progressive glucose intolerance, glycosuria, hyperglycemia, and severe body weight loss by 12 months of age. They also exhibited neuronal abnormality such as axonal degeneration and disorganization of the myelin. The phenotype of these KO rats indicates that they have the core symptoms of WS.

Leptin is a cytokine-like hormone principally produced by white adipose tissues. Defects in leptin production cause severe obesity. Leptin receptor (*Lepr*) encoded by the diabetes (*db*) gene is highly expressed in the choroid plexus. Bao et al. [59] produced *Lepr* KO rats through the CRISPR/Cas9 technology. The resulting KO rats exhibited obesity, hyperphagia, hyperglycemia, glucose intolerance, hyperinsulinemia, and dyslipidemia. In contrast, Chen et al. [79] generated *Lepr* KO rats through the TALEN technology. These rat models could complement the existing models (*db*/*db* mice and Zucker rats) [215,216] and be useful for biomedical and pharmacological research in obesity and diabetes.

Angiopoietin-like protein 8 (ANGPTL8) is a liver- and adipocyte-derived protein that controls plasma triglyceride levels. Izumi et al. [89] generated *Angptl8* KO rats through the CRISPR/Cas9 technology to clarify the roles of *Angptl8* in glucose and lipid metabolism. The resulting KO rats exhibited decreased body weight and fat content, associated with impaired lipogenesis in adipocytes. Izumi et al. [89] suggest that ANGPTL8 might be an important therapeutic target for obesity and dyslipidemia.

Hereditary aceruloplasminemia (HA) is a genetic disease characterized by iron accumulation in the liver and brain. The mutation of the ceruloplasmin (*Cp* gene is thought to be related to HA pathogenesis. Kenawi et al. [115] generated *Cp* KO rats through the CRISPR/Cas9 system. The *Cp* KO rats exhibited decreased iron concentration, transferrin saturation, plasma ceruloplasmin, and ferroxidase activity, which is considered essential for macrophage iron release. Thus, *Cp* KO rats can mimic the iron hepatosplenic phenotype in HA, which will form the basis to understand and treat the disease.

### 5.6. Models for Kidney Diseases

The renin (REN)-angiotensin system plays an important role in the control of blood pressure and renal function. Moreno et al. [31] produced *Ren* KO rats through ZFN-mediated GE system targeted to exon 5 of *Ren*. The resulting rats exhibited reduced body weight, lower blood pressure, and abnormal renal morphology (as exemplified by cortical interstitial fibrosis and abnormally shaped glomeruli). These results suggest the role of REN in the regulation of blood pressure and kidney function.

Primary hyperoxaluria type 1 (PH1) is an inherited disease caused by mutations in the mitochondrial localized alanine-glyoxylate aminotransferase (*Agxt*) gene, leading to abnormal metabolism of glyoxylic acid in the liver, subsequent endogenous oxalate overproduction, and deposition of oxalate in multiple organs, mainly the kidney. Patients with PH1 often suffer from recurrent urinary tract stones and finally renal failure. There is no effective treatment other than combined liver–kidney transplantation. Zheng et al. [94] produced *Agxt* KO rats as PH1 models through the CRISPR/Cas9 system. The resulting *Agxt* KO rats excreted more oxalate in the urine than WT animals and exhibited crystalluria with mild fibrosis in the kidney. These data suggest that *Agxt*-deficiency in mitochondria impairs glyoxylic acid metabolism and leads to PH1 in rats. This rat strain would be a valuable tool for developing innovative drugs and therapeutics.

Urate oxidase (uricase) encoded by *UOX* gene is a key enzyme whose disfunction causes hyperuricemia. Because of the low survival rate of *Uox*-deficient mice [217], Yu et al. [128] generated *Uox* KO rats through the CRISPR/Cas9 system targeting the exons 2 to 4 of *Uox*. The resulting *Uox* KO rats, called “Kunming-DY rats”, were apparently healthy, with more than a 95% survival up to one year. The male rats’ serum uric acid increased at levels significantly higher than those of WT rats. Kunming-DY rats exhibited histological renal changes including mild glomerular/tubular lesions, suggesting an alternative model animal to study hyperuricemia and associated diseases mimicking human conditions.

### 5.7. Models for Ophthalmology Diseases

The aryl hydrocarbon receptor (AHR) is a ligand-activated transcription factor which plays a role in the development of multiple tissues and is activated by a large number of ligands, including 2,3,7,8-tetrachlorodibenzo-p-dioxin. To examine the roles of the AHR in both normal biological development and response to environmental chemicals, Harrill et al. [40] produced *Ahr* KO rats through the ZFN technology targeting exon 2 of the *Ahr* and compared with an existing *Ahr* KO mouse model. *Ahr* KO rats, but not *Ahr* KO mice, displayed pathological alterations to the urinary tract, as exemplified by bilateral renal dilation (hydronephrosis). In contrast, abnormalities in vascular development were observed in *Ahr* KO mice, but not in rats. These findings suggest the differences in the role of AHR in tissue development, homeostasis, and toxicity between rats and mice.

### 5.8. Models for Hematological Systems

Hemophilia A is a genetic bleeding disorder resulting from factor VIII (FVIII or F8) deficiency. In preclinical hemophilia research, an animal model that reflects both the phenotype and pathology of the disease is required. Nielsen et al. [41] produced KO rats lacking detectable F8 activity through the ZFN technology targeting exon 16 of the *F8* gene. Episodes of spontaneous bleeding requiring treatments were observed in 70% of *F8* KO rats. Shi et al. [120] produced *F8* KO rats in which nearly the entire rat *F8* gene was inverted, causing translational stop six amino acids after the signal sequence, through the CRISPR/Cas9 technology to examine whether platelet F8 expression can prevent severe spontaneous bleeding in *F8* KO rats. They showed that the severe spontaneous bleeding phenotype in *F8* KO rats was successfully rescued by platelet-specific F8 expression through bone marrow cell transplantation. 

Autoimmune regulator (AIRE) deficiency in humans induces a life-threatening generalized autoimmune disease called autoimmune polyendocrinopathy–candidiasis–ectodermal dystrophy (APECED), and no curative treatments are available. Several models of AIRE-deficient mice have been generated; although they have been useful in understanding the role of AIRE in central tolerance, they do not reproduce the APECED symptoms accurately and, thus, there is still a need for an animal model displaying APECED-like disease. Ossart et al. [86] produced an *Aire* KO rat model using ZFN technology. The resulting KO rats exhibited several of the key symptoms of APECED disease, including alopecia, skin depigmentation, and nail dystrophy, which are much more pronounced than those in *Aire* KO mice and closer to manifestations in humans.

### 5.9. Others

Estrogens play pivotal roles in the development and function of many organ systems, including the reproductive system. Rumi et al. [48] generated estrogen receptor 1 (*Esr1*) KO rats using the ZFN system targeting exon 3 of *Esr1*. Both male and female *Esr1* KO rats were infertile. *Esr1* KO males had small testes with distended and dysplastic STs, whereas *Esr1* KO females possessed large polycystic ovaries, thread-like uteri, and poorly developed mammary glands. In addition, the uteri of *Esr1* KO rats failed to 17β-estradiol treatment. This rat model provides a new experimental tool for investigating the pathophysiology of estrogen action.

The forkhead box N1 (*FOXN1*) gene is known as a critical factor for the differentiation of thymic and skin epithelial cells. Goto et al. [69] generated *Foxn1* KO rats through the CRISPR/Cas9 technology. The resulting *Foxn1* KO rats exhibited thymus deficiency and incomplete hairless, which was characterized by splicing variants.

Multidrug resistance 1 (MDR1; also known as P-glycoprotein) is a key efflux transporter that plays an important role not only in the transport of endogenous and exogenous substances, but also in tumor MDR, one of the most important impediments to the effective chemotherapy of cancer. In rodents, two isoforms, *Mdr1a* and *Mdr1b*, encode MDR1. Liang et al. [112] produced DKO rats (in which both *Mdr1a* and *Mdr1b* had been disrupted) through the CRISPR/Cas9 system. Pharmacokinetic studies of digoxin, a typical substrate of MDR1, confirmed the deficiency of MDR1 in resulting KO rats. This rat model is a useful tool for studying the function of MDR1 in drug absorption, tumor MDR, and drug-target validation.

## 6. Perspective

Genome-engineering technology, especially the CRISPR/Cas9 system, has opened ways to create KO at the specific genes and targeted KI, perform gene replacement, and conditionally ablate gene expression with more ease. Furthermore, recently, CRISPR-based new tools such as use of other types of Cas9 endonucleases (i.e., *Staphylococcus aureus* Cas9 (SaCas9), *Francisella novicida* Cas9 (FnCas9), *Neisseria meningitidis* Cas9 (NmCas9), *Streptococcus thermophilus* Cas9 (St1Cas9), and *Brevibacillus laterosporus* Cas9 (BlatCas9)) and Cas12a (Cpf1) endonucleases (i.e., *Acidaminococcus sp*. Cpf1 (AsCpf1) and *Lachnospiraceae bacterium* Cpf1 (LbCpf1)) (reviewed by Nakade et al. [218]), prime editing (PE), and BE have emerged. For example, BE systems, as exemplified by ABE, have already been applied in rats [97,104]. The PE system is the latest addition to the CRISPR genome-engineering toolkit and represents a novel approach to expand the scope of donor-free precise DNA editing [219]. PE-based manipulation of mouse genome has already been successfully performed [220]. This approach will soon be applied to modify the rat genome. Manipulation of mtDNA by DdCBE coupled with the PB transposon system has become possible in rats [153]. These new technologies, together with the newly developed gene delivery tools such as in vitro or in vivo EP-based gene delivery (TAKE and *i*-GONAD/rGONAD), will accelerate the creation of more GM rats, which will shed light on the new role of the previously known genes, whose function had been investigated using mouse models but was judged as being far from the function inferred from human diseases.

## 7. Conclusions

To date, tools for modifying the mouse genome as exemplified by traditional gene-targeting-based KO and KI technologies have been extensively employed in mice and have proven useful for understanding the molecular mechanism underlying human diseases. However, despite these efforts, the consequences obtained from the KO or KI mice were often different from those observed in human diseases. According to Barbaric et al. [221], discrepancies between the symptoms of human diseases and mouse phenotypes may be ascribed to redundant gene networks or alternative pathways or modifiers. This point was further strengthened when a number of GM rats were successfully produced through newly developed GE technologies (see Table 1). For example, *Dmd* KO rats (but not mice) presented cardiovascular alterations close to those observed in humans, which are the main cause of death in patients. Furthermore, *Fah* KO rats developed remarkable liver fibrosis and cirrhosis, which have not been observed in *Fah* mutant mice or pigs. These findings encourage the speculation that rats may better mimic the human situation than mice.

## Figures and Tables

**Figure 1 ijms-23-02548-f001:**
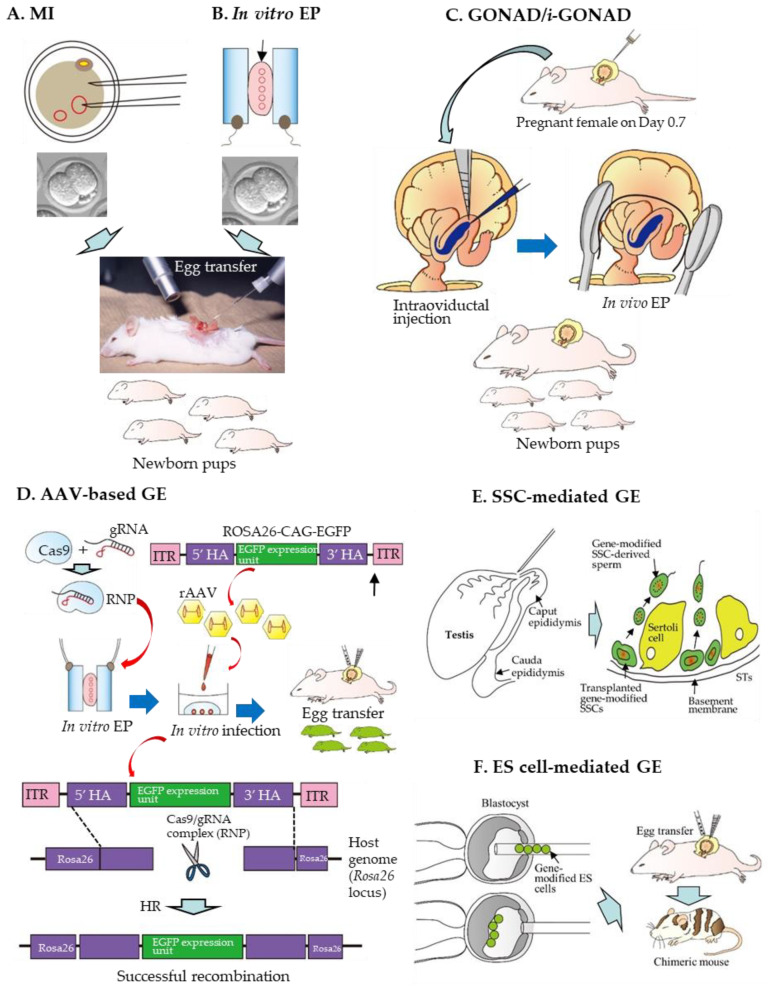
Schematic illustration of genome-edited (GE) rat production through microinjection (MI) (**A**), in vitro electroporation (EP) (**B**), genome-editing via oviductal nucleic acids delivery (GONAD) (or *improved* genome-editing via oviductal nucleic acids delivery (*i*-GONAD)) (**C**), adeno-associated virus (AAV)-based GONAD, a technique enabling in vitro viral infection of zygotes with recombinant AAV (rAAV) (**D**), spermatogonial stem cell (SCC)-mediated transgenesis (**E**), and embryonic stem (ES) cell-mediated transgenesis (**F**). This figure was drawn in-house, based on the data shown in the paper of Sato et al. [158].

**Figure 2 ijms-23-02548-f002:**
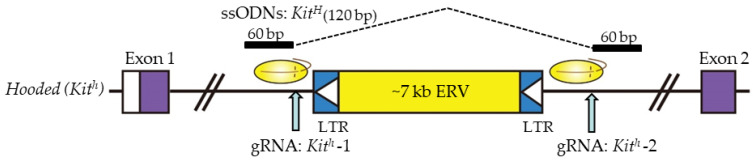
Schematic illustration of single-stranded oligodeoxynucleotide (ssODN)-based deletion of a large fragment spanning a 7,098-bp endogenous retrovirus (ERV) element within the first intron of the *Kit* gene. This figure was drawn in-house, based on the data shown in the paper of Yoshimi et al. [46].

**Figure 3 ijms-23-02548-f003:**
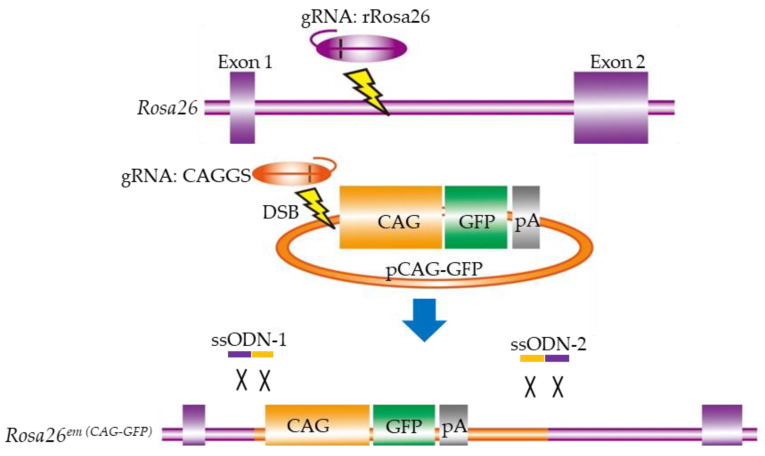
Schematic representation of the two-hit by gRNA and two-oligo with plasmid (2H2OP) method for the production of CAG-GFP knock-in (KI) rats generated using the CRISPR/Cas9 system. In the first step, Cas9, together with two gRNAs targeting the rat *Rosa26* locus and the CAG promoter in the GFP plasmids, cuts the target sites. In the second step, two ssODNs ligate each cut end to join the genomic DNA and the plasmid DNA via HDR. This figure was drawn in-house, based on the data shown in the paper of Yoshimi et al. [64]. Abbreviations: CAG, chicken β-actin-based promoter; GFP, green fluorescent protein; KI, knock-in; ssODNs, single-stranded oligodeoxynucleotides; pA, poly(A) sites.

**Figure 4 ijms-23-02548-f004:**
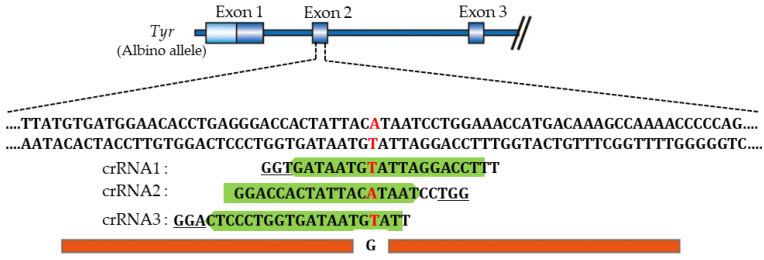
Schematic representation of knock-in (KI) experiment in rats toward the mutated *Tyr* locus performed by Aoshima et al. [141]. The target sequence (exon 2 of *Tyr*) recognized by crRNA1, 2, and 3 is shown in green. The PAM sequences are underlined. Single-stranded oligodeoxynucleotide (ssODN) (containing wild-type nucleotide “G” that corresponds to mutated nucleotide “A”) is shown in orange below the target sequence. The nucleotide “A/T” marked in red is the mutation causative of the albino phenotype. This figure was drawn in-house, based on the data shown in the paper of Aoshima et al. [141].

**Table 1 ijms-23-02548-t001:** Summary of the production of genome-edited (GE) rats from 2009 to 2021.

Method for Gene Modification	GE Tool (Mode for Gene Modification)	Rat Strain	Outcome	Target Gene	References
MI DNA or mRNA	ZFNs (indels)	SS FHH SD	Knockout (KO) rats first produced using ZFN technology with high frequency (25 to 100% disruption); showing successful germ-line transmission.	*IgM* *Rab38*	Geurts et al. (2009) [28]
MI mRNA	ZFNs (indels)	F344/Stm TM/Kyo	KO rats generated as a model for X-linked severe combined immunodeficiency (X-SCID) with rates greater than 20%; showing successful germ-line transmission.	*Il2rg*	Mashimo et al. (2010) [29]
MI mRNA	ZFNs (indels)	SD	KO rats generated to show the absence of antibody-mediated hyperacute allograft rejection; can be used for biomedical research.	*IgM*	Ménoret et al. (2010) [30]
MI mRNA	ZFNs (indels)	SS	Renin-angiotensin system-related KO rats first produced.	*Ren*	Moreno et al. (2011) [31]
MI mRNA or plasmid DNA	TALENs (indels)	Unknown	KO rats first produced using TALEN technology.	*IgM*	Tesson et al. (2011) [32]
MI mRNA/plasmid DNA	ZFN (KI)	SD LEH	Knock-in (KI) rats first produced using ZFN technology; showing successful germ-line transmission of KI phenotype.	*Mdr1a* *Pxr*	Cui et al. (2011) [33]
MI mRNA	ZFNs (indels)	F344/Stm WI	KO rats lacking either DNA-dependent protein kinase catalytic subunit (*Prkdc*) or *Prkdc* and interleukin 2 receptor gamma (*Il2rg*) generated; double knockout (DKO) F344-scid gamma (FSG) rats showed more immunocompromised phenotypes, such as the abolishment of natural killer cells.	*Prkdc* *Il2rg*	Mashimo et al. (2012) [34]
MI plasmid DNA	Engineered meganuclease	SD	KO immunodeficient rats first produced using meganucleases; showing decreased numbers of immature and mature T and B lymphocytes and normal natural killer cells.	*Rag1*	Ménoret et al. (2013) [35]
MI mRNA/plasmid DNA	ZFNs (KI)	SD	*L**oxP*-flanked (floxed) rats first produced using ZFN technology; can be useful for Cre-dependent gene disruption in vivo.	*Grin1* *Crhr1* *Tp53*	Brown et al. (2013) [36]
MI mRNA/sgRNAs	CRISPR/Cas9 (indels)	SD	KO rats first produced using CRISPR/Cas9 technology with multiple gene mutations (melanocortin 3 receptor (*Mc3r*) and melanocortin 4 receptor (*Mc4r*)) in a germ-line competent manner.	*Mc3r* *Mc4r*	Li D et al. (2013) [37]
MI mRNA/sgRNAs	CRISPR/Cas9 (indels)	SD	KO rats first produced using CRISPR/Cas9 technology with multiple gene mutations (Tet methylcytosine dioxygenase 1 (*Tet1*), Tet methylcytosine dioxygenase 2 (*Tet2*) and Tet methylcytosine dioxygenase 3 (*Tet3*)) in a germ-line competent manner.	*Tet1* *Tet2* *Tet3*	Li W et al. (2013) [38]
MI mRNAs	TALENs (indels)	WI	KO rats generated with a markedly attenuated response to a lipopolysaccharide challenge; can be used as a model for studying ethanol action and general inflammatory conditions including septic shock.	*Tlr4*	Ferguson et al. (2013) [39]
MI mRNA	ZFNs (indels)	SD	KO rats generated to examine the roles of aryl hydrocarbon receptor (*A**hr*); showing alterations to the urinary tract, including hydronephrosis and hydroureter; suggesting a role of *A**hr* in tissue development, homeostasis, and toxicity.	*A* *hr*	Harrill et al. (2013) [40]
MI mRNA	ZFNs (indels)	SD	KO rats generated as a model for hemophilia A; showing a prolonged activated partial thromboplastin time and clot formation time; can be useful for the study to treat hemophilia A.	*F8*	Nielsen et al. (2013) [41]
MI mRNA	ZFNs (indels)	Unknown	KO rat lines generated as a model for autism spectrum disorder (ASD); showing abnormalities in ASD-relevant phenotypes including juvenile play, perseverative behaviors, and sensorimotor gating.	*Fmrp* *Nlgn3*	Hamilton et al. (2013) [42]
In vitro EP mRNA/gRNA	ZFN (indels) TALEN (indels) CRISPR/Cas9 (indels)	F344/Stm	KO rats first produced using in vitro electroporation (EP) (called “Technique for Animal Knockout system by Electroporation (TAKE)”); in vitro EP resulted in an embryonic survival rate (91%) and a GE rate (73%).	*Il2rg*	Kaneko et al. (2014) [43]
MI mRNA/DNA donor	TALENs (KI)	SD	Homology-directed repair (HDR)-modified KI rats generated with multiple gene mutations with high efficiency (0.62%–5.13%).	*Hprt1* *Rosa26* *Ighm*	Remy et al. (2014) [44]
MI mRNA/sgRNAs	CRISPR/Cas9 (KI)	SD	KI rats first generated using CRISPR/Cas9-based two-cut strategy with an efficiency up to 54%.	*Trdmt1* *Nes* *Cck*	Ma et al. (2014) [45]
MI mRNA/sgRNAs	CRISPR/Cas9 (indels, KI)	F344/Stm DA PVG	Using single-stranded oligodeoxynucleotide (ssODN) donor as templates, three recessive phenotypes (including the albino phenotype by single-nucleotide polymorphism (SNP) exchange, the non-agouti phenotype by integration of a 19-bp DNA fragment, and the hooded phenotype by removal of a retrovirus-derived insertional DNA fragment) were successfully corrected.	*Tyr* *Asip* *Kit*	Yoshimi et al. (2014) [46]
MI mRNA/sgRNAs	CRISPR/Cas9 (indels)	SD	KO rats generated with multiple gene mutations (four genes) in a germ-line competent manner.	*ApoE* *B2m* *Prf1* *Prkdc*	Ma et al. (2014) [47]
MI mRNAs	ZFNs (indels)	SD	KO rats generated to examine the function of estrogen receptor 1 (*Esr1*); showing infertility, due to small testes in males, and large polycystic ovaries, thread-like uteri, and poorly developed mammary glands in females; can be a new tool for investigating the pathophysiology of estrogen action.	*Esr1*	Rumi et al. (2014) [48]
MI mRNAs/DNA donor	TALENs (KI)	SD	KI rats (carrying a pA476T mutation) first produced using TALEN technology with an efficiency of 17%.	*Nr3c1*	Ponce de Leon et al. (2014) [49]
MI mRNA/gRNAs	CRISPR/Cas9 (indels)	WI-IM	KO rats generated as a model for Duchenne muscular dystrophy (DMD); showing a decline in muscle strength, and the emergence of degenerative/regenerative phenotypes in the skeletal muscle, heart, and diaphragm; can be useful for developing therapeutic methods to treat DMD.	*Dmd*	Nakamura et al. (2014) [50]
MI mRNA	TALENs (indels)	SD	KO rats generated as a model for DMD; showing reduction in muscle strength and a decrease in spontaneous motor activity with dilated cardiomyopathy; can be useful as an animal model of DMD.	*Dmd*	Larcher et al. (2014) [51]
MI mRNA	ZFNs (indels)	SD	KO rats generated as a model for cystic fibrosis (CF); showing abnormalities in the ileum and increased intracellular mucus in the proximal nasal septa as well as reduced airway surface liquid and periciliary liquid depth; recapitulating many aspects of CF disease.	*Cftr*	Tuggle et al. (2014) [52]
MI mRNA	ZFNs (indels)	SD	KO rats generated to know that the bone morphogenetic protein receptor 2 (*Bmpr2*) mutations are linked to pulmonary arterial hypertension (PAH); displaying an intense pulmonary vascular remodeling at 3 months; suggesting that endothelial-to-mesenchymal transition (EndoMT) is linked to alterations in BPMR2 signaling.	*Bmpr2*	Ranchoux et al. (2015) [53]
In vitro EP mRNA/gRNA/ssODN	CRISPR/Cas9 (indels, KI)	WI	KI and KO rats produced through in vitro EP in the presence of CRISPR/Cas9 components with efficiencies of 33% and 88%, respectively.	*Il2rg*	Kaneko and Mashimo (2015) [54]
MI mRNAs	TALENs (indels)	Unknown	KO rats generated to examine the function of cold-inducible RNA-binding protein *(**Cirp*) in heart; CIRP modulates cardiac repolarization by negatively adjusting the expression and function of Ito channels.	*Cirp*	Li et al. (2015) [55]
MI mRNAs	TALENs (indels)	SD	KO rats displayed typical dyslipidemia, although no obvious atherosclerotic lesion was noted in the enface aortas and aortic root; can be a novel model for dyslipidemia and is used in the research of atherosclerosis.	*ApoE*	Wei et al. (2015) [56]
MI protein/gRNA	CRISPR/Cas9 (KI) TALENs (KI)	SD	Cas9 protein was more efficient at HDR than *Cas9* mRNA, while TALEN protein was less efficient than TALEN mRNA for inducing HDR.	*Rosa26* *Foxp3* *Anks3*	Ménoret et al. (2015) [57]
MI protein/sgRNA/DNA donor	CRISPR/Cas9 (KI)	SD	A DNA cassette (composed of a green fluorescent protein (*GFP*) reporter sequence flanked by the two pairs of *lox* sites) was successfully inserted in the reverse orientation into the target (leucine-rich repeat-containing G-protein coupled receptor 5 (*Lgr5***)**) locus, which was later inverted by Cre-mediated recombination.	*Lgr5*	Wang et al. (2015) [58]
ES cell-mediated GE plasmid DNA	CRISPR/Cas9 (KI)	ES cells	The first production of KI rats using CRISPR/Cas9-engineered embryonic stem (ES) cells; in vitro EP of Cas9 plasmid, sgRNA plasmid, and a KI vector led to enhanced homologous recombination (HR) efficiency up to 36%.	*Kat II*	Yamamoto et al. (2015) [14]
MI mRNA/gRNA	CRISPR/Cas9 (indels)	SD	KO rats generated as a model for obesity; showing obesity, hyperphagia, hyperglycemia, glucose intolerance, hyperinsulinemia, and dyslipidemia, as well as a decrease in bone volume and bone mineral density of the femur; can be useful for the research on obesity and diabetes.	*L* *e* *pr*	Bao et al. (2015) [59]
SSC-mediated GE plasmid DNA	CRISPR/Cas9 (indels)	SD-derived SSCs	The first creation of spermatogonial stem cell (SSC)-derived GE KO rats; when recipient male rats were transplanted with engineered SSCs into seminiferous tubules (STs), and then mated with wild-type (WT) females, the resulting progeny harbored indels with ~10% efficiency.	*Epsti1* *Erbb3*	Chapman et al. (2015) [60]
MI mRNA/sgRNA	CRISPR/Cas9 (indels)	SD	KO rats generated as a model for Rett syndrome (RS); showing body weight loss, anxiety tendency, and cognitive deficits; recapitulating the major symptoms of RS patients.	*Mecp2*	Zhai et al. (2016) [61]
MI mRNA	ZFNs (indels)	SD	KO rats generated as a model for RS; showing behavioral and motor deficits in male and female rats; can be used in RS research.	*Mecp2*	Patterson et al. (2016) [62]
MI mRNA/sgRNA	CRISPR/Cas9 (indels)	SD	KO rats generated as a model for hereditary tyrosinemia type I (HT1); showing major phenotypic manifestations of human HT1, including hypertyrosinemia, liver failure, and renal tubular damage; also showing remarkable liver fibrosis and cirrhosis; can be used as a model of HT1 with liver cirrhosis.	Fah	Zhang et al. (2016) [63]
MI mRNA/gRNA/ lssODN	CRISPR/Cas9 (KI)	F344/Stm	The ssODN-mediated KI rats were successfully generated through two-hit by gRNA and two-oligo with plasmid (2H2OP) system; can be applied to any target site with any donor vector without the need to construct homology arms, thus simplifying genome engineering in living organisms.	*Tyr* *Thy1* *Rosa26* *Cyp2d* *Sirpa*	Yoshimi et al. (2016) [64]
MI mRNA/gRNA/ ssODN	CRISPR/Cas9 (KI)	SD	Usage of chemically modified (phosphorothioate-modified) ssODN donors is shown to be effective in improving KI efficiency.	*Cftr*	Renaud et al. (2016) [65]
MI mRNA	TALENs (indels)	SD	Homozygous KO rats are embryonically lethal; both male and female fumarate hydratase (*Fh*)+/- KO rats exhibited reduced litter size; also showing hematopoietic and kidney dysfunction; *Fh*+/- KO rats can be useful for further functional fumarate hydratase studies.	*Fh*	Yu et al. (2016) [66]
MI mRNA	TALENs (indels)	F344	Homozygous KO rats are embryonically lethal; DEP domain-containing 5 (*Depdc5*)(+/-) rats developed normally and exhibited no spontaneous electroclinical seizures, but had altered cortical neuron excitability and firing patterns; can be a model for mTORopathy.	*Depdc5*	Marsan et al. (2016) [67]
MI protein (or mRNA)/sgRNA/donor plasmid	CRISPR/Cas9 (KI)	SD	Both Scr7 (non-homologous end joining (NHEJ) inhibitor) and Cas9 protein were used to increase KI efficiency; consequently, this combination successfully increased the HR-mediated precise modification.	*Fabp2* *Dbndd1*	Ma et al. (2016) [68]
MI mRNA/sgRNAs	CRISPR/Cas9 (indels)	WI	KO rats generated to examine the function of forkhead box N1 (*Foxn1*) that is known to a critical factor for differentiation of thymic and skin epithelial cells; showing thymus deficiency and incomplete hairless.	*Foxn1*	Goto et al. (2016) [69]
MI mRNA/sgRNAs	CRISPR/Cas9 (indels)	SD	*Mc3r* and *Mc4r* single-KO or DKO rats generated to examine *Mc3r* and *Mc4r* more deeply; *Mc3r* KO rats displayed hypophagia and decreased body weight, while *Mc4r* KO and DKO exhibited hyperphagia and increased body weight; both *Mc4r* KO and DKO are good models for obesity and diabetes research.	*Mc3r Mc4r*	You et al. (2016) [70]
MI mRNA	TALENs (indels)	SD	KO rats generated to evaluate the role of reduced nuclear factor (erythroid-derived 2)-like-2 (NRF2)-regulated antioxidant defenses; suppression of NRF2 antioxidant defenses plays a role in the development of salt-induced oxidant stress, endothelial dysfunction, and microvessel rarefaction; can be used as therapeutics to ameliorate vascular oxidant stress in humans.	*Nrf2*	Priestley et al. (2016) [71]
MI mRNA	ZFNs (indels)	F344/Stm	KO rats generated to evaluate the role of reduced NRF2-regulated antioxidant defenses were sensitive to aflatoxin B1 toxicity; can be a new model animal in toxicology.	*Nrf2*	Taguchi et al. (2016) [72]
MI mRNA/gRNA	CRISPR/Cas9 (indels)	Unknown	KO rats generated to investigate the functions of cytochrome P450 2E1 (CYP2E1) were viable and fertile and did not display any obvious physiological abnormities; can be a powerful tool for the study of CYP2E1 in the chemical metabolism, toxicity, carcinogenicity.	*Cyp2e1*	Wang et al. (2016) [73]
MI mRNA/gRNA	CRISPR/Cas9 (indels)	WI	KO rats generated to examine the function of Mohawk homeobox (*Mkx*) in tenogenesis; showing systemic hypoplasia of tendons and earlier heterotopic ossification of the Achilles tendon; can be used for studying tendon physiology and tissue engineering.	*Mkx*	Suzuki et al. (2016) [74]
MI mRNA	ZFNs (indels)	SD	KO rats generated as a model for pseudoxanthoma elasticum, a heritable ectopic mineralization disorder; showing reduced levels of plasma inorganic pyrophosphate (PPi), suggesting a critical role of hepatic ATP binding cassette subtype C number 6 (ABCC6) in contributing to plasma PPi levels.	*Abcc* *6*	Li et al. (2017) [75]
MI mRNA	ZFNs (indels)	SD	KO rats generated as model of Wolfram syndrome (WS), a rare autosomal-recessive disorder characterized by juvenile-onset diabetes, optic atrophy, and hearing loss; exhibiting the core symptoms of WS.	*Wfs1*	Plaas et al. (2017) [76]
MI mRNA/sgRNA	CRISPR/Cas9 (indels)	SD	KO rats specifically impairs long-term synaptic plasticity and hippocampus-dependent learning in a manner resembling the key symptoms of Fragile X syndrome (FXS).	*Fmr1*	Tian et al. (2017) [77]
MI mRNA/gRNA	CRISPR/Cas9 (indels)	Unknown	KO rats generated to investigate the role of myosin light-chain 4 (MYL4) in atrial cardiomyopathy; showing progressive atrial cardiomyopathy; suggesting that *Myl4* is as a key gene required for atrial contractile, electrical, and structural integrity.	*M* *yl* *4*	Peng et al. (2017) [78]
MI mRNA	TALENs (indels)	SD	KO rats generated as obese rat model; showing early onset of obesity and infertility.	*Lepr*	Chen et al. (2017) [79]
MI mRNA/gRNA	CRISPR/Cas9 (indels)	SD	KO rats generated to determine the function of patatin-like phospholipase domain containing 5 (*P**npla5*); were viable, but showed a variety of abnormalities related to lipid metabolism; can be a possibly model for the treatment of cardiovascular diseases.	*P* *npla5*	Liu et al. (2017) [80]
MI mRNA	ZFNs (indels)	SD	KO rats generated as a model for Phelan–McDermid syndrome (PMS); exhibit impaired long-term social recognition memory and attention, and reduced synaptic plasticity in the hippocampal-medial prefrontal cortex pathway.	*Shank3*	Harony-Nicolas et al. (2017) [81]
MI mRNA/gRNA (or mRNA/gRNA/ssODN)	CRISPR/Cas9 (KI, indels)	SS	GM rats generated to assess the candidacy of a novel predicted long non-coding RNA (lncRNA) showing strong association of cardiac QT-interval; KO rats showed aberrant, short QT-intervals, and elevated blood pressure; KI rats showed the rescued phenotype.	*Rffl-lnc1*	Cheng et al. (2017) [82]
MI mRNA	TALENs (indels)	SD	KO rats generated to examine why rats are naturally resistant to *Schistosoma japonicum* infection; high expression levels of inducible nitric oxide synthase (*iNOS*) in rats play an important role in blocking the egg-induced granuloma formation of *S. japonicum*; providing insights for understanding the pathogenesis of human schistosomiasis.	*iNOS*	Shen et al. (2017) [83]
MI mRNA	ZFNs (indels)	SS	KO rats generated to define the importance of Kir5.1 (encoded by potassium inwardly-rectifying channel, subfamily J, member 16 (*Kcnj16*)) in blood pressure control under salt-induced hypertension; exhibiting hypokalemia and reduced blood pressure, and 100% mortality within a few days triggered by salt wasting; demonstrating that Kir5.1 channels are key regulators of renal salt handling.	*Kcnj16*	Palygin et al. (2017) [84]
ES cell-mediated GE plasmid	CRISPR/Cas9 (KI)	ES cells	KI rats derived from the CRISPR/Cas9-engineered ES cells; two clones showing non-disruptive KI of *Discosoma* coral red fluorescent protein (*dsRed*) at the Sry-box transcription factor 10 (*Sox10*) locus produced germ-line chimeras; showing expression of *dsRed* in the *Sox10*-expressing cells (oligodendrocyte lineage cells) in vivo.	*Sox10*	Chen et al. (2017) [15]
MI mRNA/gRNA	CRISPR/Cas9 (indels)	SD	DKO rats generated to investigate cytochrome P450 3A (CYP3A) functions; were viable and fertile, and had no obvious physiological abnormities; can be a powerful tool for the study of the physiological and pharmacological roles of CYP3A, especially in drug and chemical metabolism in vivo.	*Cyp3a1* *Cyp3a2*	Lu et al. (2017) [85]
MI mRNA	ZFNs (indels)	BN	KO rats generated as a model for autoimmune polyendocrinopathy-candidiasis-ectodermal dystrophy (APECED); displaying several of the key symptoms of APECED, including alopecia, skin depigmentation, and nail dystrophy	*Aire*	Ossart et al. (2018) [86]
MI mRNA/gRNA/ssODN	CRISPR/Cas9 (KI)	SD	KI rats (carrying a Fused in sarcoma (*Fus*)-point mutation (R521C)) generated as a model for amyotrophic lateral sclerosis (ALS); developing adult-onset learning and memory behavioral deficits cognitive impairment; suggesting a new role of FUS in sleep and circadian regulation.	*Fus*	Zhang et al. (2018) [87]
MI mRNA	TALENs (indels)	SD	Immunodeficient rats with DKO rats generated for obtaining more profound immunosuppressed phenotype; can be useful as recipients for long-term studies concerning tissue humanization of different tissues.	*Il2rg* *Rag1*	Ménoret et al. (2018) [88]
MI mRNA/gRNA	CRISPR/Cas9 (indels)	F344/Stm	KO rats generated to clarify the roles of angiopoietin-like protein 8 (ANGPTL8) in glucose and lipid metabolism; showing lower body weight and fat content, associated with impaired lipogenesis in adipocytes; can be an important therapeutic target for obesity and dyslipidemia.	*Angptl8*	Izumi et al. (2018) [89]
*i*-GONAD protein/gRNA (or protein/gRNA/ssODN)	CRISPR/Cas9 (indels, KI)	SD LEW BN	*Improved* GONAD (*i*-GONAD) was found useful for producing KO and KI rats with efficiencies of 62 to 97% and 9%, respectively.	*Tyr* *Sox16*	Takabayashi et al. (2018) [90]
*i*-GONAD (rGONAD) protein/gRNA (or protein/gRNA/ssODN)	CRISPR/Cas9 (indels, KI)	WKY DA	Most suitable condition for in vivo gene delivery towards rat preimplantation embryos was assessed; *i*-GONAD was found useful for producing KO and KI rats with relative high efficiency.	*Tyr*	Kobayashi et al. (2018) [91]
MI mRNA/gRNA	CRISPR/Cas9 (indels)	Unknown	DKO rats (lacking both apolipoprotein E (*ApoE*) and low-density lipoprotein receptor (*Ldlr*)) generated as a model for familial hypercholesterolemia; showing hypercholesterolemia, hepatosteatosis, and atherosclerosis.	*ApoE* *Ldlr*	Zhao et al. (2018) [92]
MI mRNA/gRNA	CRISPR/Cas9 (indels)	SD	KO rats generated to investigate whether Cas9 nickase (Cas9n)-mediated GE can efficiently correct the HT1; tail-vein injection of adenoviral vectors carrying Cas9n, repair donor template, and sgRNA successfully corrected the phenotype of fumarylacetoacetate hydrolase (*Fah*) KO rats.	*Fah*	Shao et al. (2018) [93]
MI mRNA/gRNA	CRISPR/Cas9 (indels)	Unknown	KO rats generated as a model for primary hyperoxaluria type 1 (PH1); showing crystalluria with abnormal phenotypes, such as a slight dilatation of renal tubules with mild fibrosis in the kidney; suggesting impairment of glyoxylic acid metabolism; can be a useful tool for the development and evaluation of drugs and therapeutics.	*Agxt*	Zheng et al. (2018) [94]
In vitro EP (protein/gRNAs/lssDNA)	CRISPR/Cas9 (KI)	F344	Efficient approaches enabling the quick generation of floxed alleles in mice and rats were taken using in vitro EP in the presence of Cas9 protein, gRNAs, and long single-stranded DNA (lssDNA); homozygous KI oocytes carrying tissue-specific Cre gene were successfully obtained with high efficiency.	*Vapb*	Miyasaka et al. (2018) [95]
MI mRNA/sgRNA	CRISPR/Cas9 (indels)	SD	KO rats generated as a model for RNase T2 deficiency; showing no evidence of cystic lesions, but hippocampal neuroinflammation, altered lysosomal function, and cognitive defects; can be a useful model to study the RNase T2 function.	*RNase T2*	Sinkevicius et al. (2018) [96]
MI mRNA/sgRNA	ABEs (indels)	Unknown	KO rats generated as a model for Glycogen storage disease type II (GSDII) with an efficiency of 85%; the adenine base editor (ABE) system is a powerful and convenient tool to introduce precise base conversions in rodents.	*Gaa*	Yang et al. (2018) [97]
MI plasmid DNA	CRISPR/Cas9 (indels)	DA	KO rats generated as a model for Fabry disease (FD); showing substantial serum and tissue accumulation of α-galactosyl glycosphingolipids and had pronounced mechanical pain behavior.	*α-Gal A*	Miller et al. (2018) [98]
MI mRNA	TALENs (indels)	SD	Oxoglutarate dehydrogenase (*Ogdh*)−/− rats are lethal; *Ogdh*+/− rats had higher body weight; treatment of *Ogdh*+/− rats with a high-fat diet resulted in liver dysfunction; *Ogdh*+/− rats can be used as model for investigations of metabolic syndrome and obesity-related hepatic carcinogenesis.	*Ogdh*	Fan et al. (2018) [99]
MI mRNA	ZFNs (indels)	WI	KO rats developed normally, but exhibited spontaneous locomotor hyperactivity and cognitive dysfunctions; can be a model for human diseases involving aberrant dopamine functions.	*Dat*	Leo et al. (2018) [100]
MI plasmid	CRISPR/Cas9 (indels)	F344 SD	KO rats generated to know the role of complement in chemotherapy-induced peripheral neuropathy (CIPN); suggesting that complement may be a new target for the development of novel therapeutics to manage CIPN.	*C3*	Xu et al. (2018) [101]
SSC-mediated GE plasmid	TALENs (indels)	SD-derived SSCs	SSCs-derived GE KO rats showed immunodeficiency with lacked mature B and T cells; allowing growth of human tumors; can be used for xenograft studies.	*Rag2*	Noto et al. (2018) [102]
MI mRNA/sgRNAs	CRISPR/Cas9 (indels)	Unknown	Multiple-gene KO rats generated to examine the function of defensin beta (*Defb*) expressed abundantly in the epididymis; showing subfertility and decreased motility of sperm; suggesting that *Defb* family members affect sperm maturation by a synergistic pattern in the epididymis.	*Defb23 Defb26 Defb42*	Zhang et al. (2018) [103]
MI mRNA/sgRNA	ABE (indels)	SD	For hemogen (*Hemgn*) targeting, out of 15 rats obtained, 14 contained an A to G conversion at the 14th base distal from the protospacer adjacent motif (PAM), indicating the high base-editing efficiency.	*Hemgn* *Ndst* *1* *Ndst4*	Ma et al. (2018) [104]
MI mRNA/sgRNA	CRISPR/Cas9 (indels)	SD	KO rats generated to assess the roles of cytochrome P450, subfamily 2, polypeptide 11 (CYP2C11); were viable and had no obvious abnormalities, with the exception of reduced fertility; can be a valuable tool to study the in vivo function of CYP2C11.	*Cyp2c11*	Wei et al. (2018) [105]
MI mRNA/gRNA (or mRNA/gRNA ssODN)	CRISPR/Cas9 (indels, KI)	Unknown	KI (carrying F508del mutation) and KO rats generated as a model for CF; both rats exhibit CF phenotypic anomalies such as vas deferens agenesis and tooth enamel defects; can be used as a novel resource to advance the development of CF therapeutics.	*Cftr*	Dreano et al. (2019) [106]
MI plasmid DNA	CRISPR/Cas9 (indels)	DA	KO rats generated as a model for FD; showing pronounced renal tubule dysfunction and mitral valve thickening; can be further used to study disease mechanisms and test therapies.	*α-Gal A*	Miller et al. (2019) [107]
MI mRNA/sgRNA	CRISPR/Cas9 (indels)	SD	KO rats generated a model for ASDs; showing impaired social memory (but not impaired social interaction behaviors), impaired learning and memory, increased anxiety-like behavior, increased mechanical pain threshold, and decreased thermal sensation.	*Shank3*	Song et al. (2019) [108]
MI mRNA/sgRNA	CRISPR/Cas9 (indels)	SD	KO rats generated as a model for pulmonary hypertension (PH); showing PH pathogenesis associated with low serum-albumin concentration; can be used for understanding the mechanisms of PH and testing therapeutic molecules.	*Kcnk3*	Lambert et al. (2019) [109]
MI mRNA/sgRNA	CRISPR-Cpf1 (indels)	SD	ApoE KO rats displayed hyperlipidemia and aortic lesions; showing that the Cpf1 system can target single or multiple genes efficiently; these rats can be helpful for understanding initial-stage atherosclerosis.	*ApoE* *Ldlr*	Lee et al. (2019) [110]
In vitro EP (protein/sgRNA/plasmid DNA) rAAV	CRISPR/Cas9 (KI, indels)	BN	Superovulation is successfully conducted to Brown Norway (BN), which is a very difficult rat strain to superovulate; in vitro EP or infection with recombinant adeno-associated virus (rAAV) enables KO or KI in the in vitro fertilized (IVF) rat embryos efficiently.	*Tyr*	Honda et al. (2019) [111]
MI mRNA/sgRNA	CRISPR/Cas9 (indels)	SD	KO rats exhibited complete loss of multidrug resistance protein 1 (MDR1) in the liver, small intestine, brain, and kidney; can be useful for studying the function of MDR1 in drug absorption, tumor multidrug, and drug target validation.	*Mdr1a/b*	Liang et al. (2019) [112]
MI protein/sgRNA	CRISPR/Cas9 (indels)	SD	KO rats generated as a model for X-linked hydrocephalus (XLH); hemizygous male mutants developed hydrocephalus and delayed development.	*L1cam*	Emmert et al. (2019) [113]
MI mRNA/sgRNA/ssODN	CRISPR/Cas9 (KI)	SD	KI rats (carrying a R571X mutation) generated to mimic Cockayne syndrome (CS); showing atrophy and dysmyelination in the cerebellar cortex; RNA-seq suggests that transcription dysregulation could contribute to the CS pathogenesis.	*Csb*	Xu et al. (2019) [114]
MI protein/sgRNA	CRISPR/Cas9 (indels)	SD	KO rats generated to understand the mechanisms leading to iron excess in HA; showing decreased iron concentration and transferrin saturation; suggesting that the reported role of ceruloplasmin (*Cp*) cannot fully explain the iron hepatosplenic phenotype in hereditary aceruloplasminemia (HA).	*Cp*	Kenawi et al. (2019) [115]
MI mRNA	ZFNs	SD	KO rats generated as a model for PAH; showing age-dependent spontaneous PAH with a low penetrance (16%-27%), similar to that in humans; can be a promising tool to study the pathogenesis of PAH.	*Bmpr2*	Hautefort et al. (2019) [116]
In vitro EP protein/gRNA	CRISPR/Cas9 (indels)	SD	KO rats generated to examine the role of 5-hydroxytryptamine receptor 7 (*Htr7*) in blood pressure regulation; neither the male nor female KO rats were different in body size, fat weights, or mass of organs (kidney, heart, and brain) important to blood pressure; can be used to investigate the importance of *Htr7* in blood pressure regulation.	*Htr7*	Demireva et al. (2019) [117]
MI mRNA/two sgRNAs/donor plasmid containing two flanking *loxP* sites	CRISPR/Cas9 (KI)	SD	KI rats generated to create a conditional glucocorticoid receptors (GR) knockdown using a dual sgRNA strategy; affording high-precision knockdown of GR across a variety of contexts, ranging from neuronal depletion to circuit-wide manipulations.	*Nr3c1*	Scheimann et al. (2019) [118]
MI mRNA/ sgRNA/donor plasmid	CRISPR/Cas9 (KI)	LE	Two KI rat lines (D1-Cre and A2a-Cre) generated for Cre-mediated gene manipulation; will be used to study both normal brain functions and neurological and psychiatric pathophysiology.	*Drd1a* *Adora2a*	Pettibone et al. (2019) [119]
MI mRNA/gRNA	CRISPR/Cas9 (indels)	SS	KO rats generated as a model for hemophilia; showing spontaneous bleeding in the soft tissue, muscles, or joints occurred in 100% of factor 8 (*F8*)-/- rats; bone marrow reconstitution rescued the spontaneous bleeding in rat HA hemophilia.	*F* *8*	Shi et al. (2020) [120]
*i*-GONAD protein/gRNAs	CRISPR/Cas9 (indels)	BN	*i*-GONAD under EP conditions with current values ranging from 150 to 200 mA enabled production of KO rats with efficiencies ranging from 75% to 24%.	*Tyr*	Takabayashi et al. (2020) [121]
MI mRNA/gRNA	CRISPR/Cas9 (indels)	SD	KO rats generated as a model for Angelman syndrome (AS); showing deficits in motor coordination as well as learning and memory, phenotypically mirroring human AS; can be a model for the study of AS.	*Ube3a*	Dodge et al. (2020) [122]
MI mRNA/gRNA	CRISPR/Cas9 (indels)	SD	KO rats generated to explore the function of complexin I (*Cplx1*); showing a complex neurobehavioral phenotype including profound ataxia, dystonia, movement and exploratory deficits, and increased anxiety and sensory deficits, but had normal cognitive function.	*Cplx1*	Xu et al. (2020) [123]
MI mRNA (or protein)/gRNAs/plasmid donor	Cas12a/Cpf1 (MAD7) (indels, KI)	SD	MAD7 is capable of generating indels, small DNA insertion, and fluorescent gene tagging of endogenous genes, such as Cas9; causing indels with 20% efficiency in rats; MAD7 can expand the CRISPR toolbox for genome engineering.	*Calb2*	Liu et al. (2020) [124]
In vitro EP mRNA/gRNA (or mRNA/gRNA/ lssDNA)	CRISPR/Cas9 (indels, KI)	LE	KO rats generated to know the role of glutamate decarboxylase 1 (GAD1) deficiency in pathogenesis of schizophrenia; showing complex behavioral changes, such as hypoactivity in a novel environment and decreased preference for social novelty; can be a model for studying the mechanism of schizophrenia.	*Gad1*	Fujihara et al. (2020) [125]
MI protein/sgRNA/ssODN	CRISPR/Cas9 (KI)	SD	KI rats (carrying a R350P mutation) generated as a model for desminopathy; showing the phenotype of desminopathy; can be a useful tool for understanding of the disease and testing therapeutic approaches to delay disease progression.	*Des*	Langer et al. (2020) [126]
MI mRNA/sgRNA/ssODN	CRISPR/Cas9 (KI)	SD	KI rats (carrying a D205N mutation) generated as a model for primary hyperoxaluria type 1 (PH1); showing hyperoxaluria at 1 month of age and exhibited severe renal calcium oxalate deposition after ethylene glycol challenge; can be a useful model for understanding the disease and developing therapeutics.	*Agxt*	Zheng et al. (2020) [127]
MI mRNA/gRNA	CRISPR/Cas9 (indels)	SD	KO rats generated a model for hyperuricemia; were apparently healthy with more than a 95% survival up to one year and showed renal abnormality such as mild glomerular/tubular lesions; can be useful for studying hyperuricemia.	*Uox*	Yu et al. (2020) [128]
MI mRNA/sgRNA (or mRNA/sgRNA/ssODN)	CRISPR/Cas9 (KI, indels)	WI	Cytochrome P450 family 27 subfamily B member 1 (*Cyp27b1*) KO, vitamin D receptor (*Vdr*) KO, and *Vdr* KI (carrying a R270L mutation) rats generated as a model for human type II rickets; showing growth retardation and abnormal bone formation; can be useful for elucidating the molecular mechanism of vitamin D action.	*Cyp27b1* *Vdr*	Nishikawa et al. (2020) [129]
MI mRNA/donor DNA	ZFNs (KI)	SD	KI rats (carrying a G551D variant) generated as a model for CF; showing that the epithelia recapitulates the expected absence of cystic fibrosis transmembrane conductance regulator (CFTR) activity, which was restored with ivacaftor, a medication used to treat CF.	*Cftr*	Birket et al. (2020) [130]
In vitro EP protein/gRNA	CRISPR/Cas9 (indels)	SD	KO rats generated to investigate the role of monocarboxylate transporter 8 (MCT8) (encoded by solute carrier family 16 member 2 coding for monocarboxylate transporter 8 (*Slc16a2*)) during spermatogenesis; showing growth delay during postnatal development with reduced sperm motility and viability; suggesting the role of MCT8 in spermatogenesis.	*Slc16a2*	Bae et al. (2020) [131]
MI mRNA	ZFNs (indels)	WI	KO rats generated to examine whether dopamine plays a key role in sexual behavior; showing more rapid acquisition of stable sexual activity levels and to higher levels of sexual motivation and activity; can be confirmed that dopamine has a key role in sexual behavior.	*D* *at*	Sanna et al. (2020) [132]
MI mRNA/sgRNA	CRISPR/Cas9 (indels)	SD	KO rats generated to enable more sophisticated modeling of pain, itch, and asthma; showing apparently normal behavioral responses to pain and itch, although transient receptor potential family member ankyrin 1 (TRPA1)-dependent immune cell infiltration into the lung was seen under the asthma-inducing condition; useful for studying that TRPA1 can be as a drug target.	*Trpa1*	Reese et al. (2020) [133]
MI mRNA/sgRNA	CRISPR/Cas9 (indels)	SD	KO rats generated to study the function of cytochrome P450 family 2 subfamily J (Cyp2J) isoform genes in vivo; were viable and fertile; can be a useful tool to study the function of *C**yp**2J* in drug metabolism and cardiovascular disease.	*Cyp2J2* *Cyp2J* *3* *Cyp2J10*	Lu et al. (2020) [134]
MI mRNA	TALENs (indels)	F344	KO rats generated to study the role of coiled-coil domain containing 85C (*Ccdc85c*) in the pathogenesis of spontaneous hydrocephalus shown in hemorrhagic *hydrocephalus* (*hhy*) mice; showing non-obstructive hydrocephalus, subcortical heterotopia, and intracranial hemorrhage; suggesting a role of *Ccdc85c* in cerebral development.	*Ccdc85c*	Konishi et al. (2020) [135]
MI mRNA/gRNA/ssODN	CRISPR/Cas9 (KI)	LE	KI rats (carrying a Psen1LF mutation) generated to determine potential early pathogenic changes caused by this mutation; survived into adulthood; showing increased levels of Aβ43, a longer and potentially more amyloidogenic Aβ form.	*Psen1*	Tambini and D’Adamio (2020) [136]
MI protein/sgRNAs/donor DNA	CRISPR/Cas9 (KI)	SD	KI rats (carrying GGGGCC (G4C2) repeats) generated as a model for ALS; showing motor deficits from 4 months of age; can be used for investigating the neurotoxicity in chromosome 9 open reading frame 72 (*C9orf72*)-related ALS.	*C9orf72*	Dong et al. (2020) [137]
MI protein/sgRNAs/donor DNA	CRISPR/Cas9 (KI)	SD	KI rats generated as a model for the Mediterranean SNP G6PDS188F; showing glucose-6-phosphate dehydrogenase (G6PD) activity, but not expression, was reduced to 20% of WT littermates; replacement of a single amino acid (S188F) in G6PD, a rate-limiting enzyme in the pentose phosphate pathway, may be related to pathogenesis of vascular diseases.	*G6pd*	Kitagawa et al. (2020) [138]
MI mRNA/sgRNA	CRISPR/Cas9 (indels)	SD	KO rats generated to examine the role of solute carrier organic anion transporter family member 1B2 (*Slco1b2*); in those rats, the serum levels of bilirubin and bile acids, the substrates of organic anion transporting polypeptide organic anion transport polypeptide 1b2 (OATP1B2), increased; can be a disease model to study hyperbilirubinemia-related diseases.	*Slco1b2*	Ma et al. (2020) [139]
MI, in vitro EP mRNA/sgRNAs/dsDNA donor vectors (or mRNA/sgRNAs/lssDNA)	CRISPR/Cas9 (KI)	F344	A new powerful method, called Combi-CRISPR, was developed for achieving plasmid-based KI in rat embryos using the CRISPR/Cas9 system; in other words, this method is the combination of highly efficient editing via NHEJ and the low-efficiency but precise editing via HDR.	*Pvalb* *Th*	Yoshimi et al. (2021) [140]
*i*-GONAD protein/gRNA/ssODN	CRISPR/Cas9 (KI)	SD	Effects of 3 gRNAs (which recognize different portions of the target locus, but also overlap each other in the target locus) and of commercially available KI-enhancing drugs on the KI efficiency; KI efficiency largely depends on the type of gRNA used; none of drugs are significantly effective for KI efficiency.	*Tyr*	Aoshima et al. (2021) [141]
MI protein/gRNA (or protein/gRNA/ssODN)	CRISPR/Cas9 (indels, KI)	BN	KO rats generated as a model for retinal dystrophy (RD); showing reduced vision and structural abnormalities, such as overall retinal thinning; can be new animal model for retinal dystrophy.	*Lrat*	Koster et al. (2021) [142]
MI mRNA/sgRNA	CRISPR/Cas9 (indels)	WI	KO rats generated to investigate cytochrome P450 family 24 subfamily A member 1 (CYP24A1)-dependent or -independent metabolism of 25(OH)D_3_, the prohormone of calcitriol; found to be valuable for metabolic studies of vitamin D and its analogs.	*Cyp24a1*	Yasuda et al. (2021) [143]
In vitro EP mRNA	TALENs (indels)	SD	KO rats generated to elucidate the functions of NK3 homeobox 1 (*Nkx3.1*); showing reduced fertility, decreased prostate weights, and increased epithelial cell layers; can be a model for studying the role of *Nkx3.1* in decreased prostate weights, fertility, and breast cancer, as well as in prostate cancer.	*N* *kx* *3.1*	Lee et al. (2021) [144]
MI mRNA/sgRNA	CRISPR/Cas9 (indels)	Unknown	KO rats generated to elucidate the roles of alpha/beta-hydrolase domain 6 (ABHD6) in vivo; showing normal appearance, but caused more frequent urination in the stimulated bladder.	*Abhd6*	Noguchi et al. (2021) [145]
MI plasmid	CRISPR/Cas9 (indels)	SS	KO rats lacking adaptor protein 1 (shc1), encodes 3 main protein isoforms (*p52Shc* and *p46Shc* isoforms) generated to further investigate the function of *Shc1*; showing severe gait abnormalities accompanied by dilated cardiomyopathy.	*p52Shc* *p46Shc*	Miller et al. (2021) [146]
MI mRNA/sgRNAs	CRISPR/Cas9 (indels)	WI	KO rats generated to investigate the in vivo roles of carnosine synthase 1 (*Carns1*); showing a significant impairment of contractile function in the cardiac muscle; suggesting a role of high-cholesterol diet (HCD) in the regulation of Ca^2+^ handling and excitation–contraction coupling of cardiac muscle.	*Carns1*	de Souza Gonçalves et al. (2021) [147]
MI mRNA/sgRNAs	CRISPR/Cas9 (indels)	SD	KO rats generated to examine the function of transient receptor potential cation channel subfamily V member 4 (*Trpv4*) in regulating osteoarthritic pain; showing suppression of joint pain under the monoiodoacetate-induced osteoarthritic pain model; suggesting that inhibition of TRPV4 might be a novel potent analgesic strategy for treating osteoarthritic pain.	*Trpv4*	Soga et al. (2021) [148]
MI mRNA/sgRNAs	CRISPR/Cas9 (indels)	SD	KO rats generated to examine the role of carboxylesterase 2A (*Ces2a*) in drug metabolism mediated by carboxylesterase 2 (CES2); showing obesity, insulin resistance, and liver fat accumulation, which are consistent with the symptoms of nonalcoholic fatty liver disease (NAFLD); can be used as a model for NAFLD.	*Ces2a*	Liu et al. (2021) [149]
MI protein/two plasmids (one containing gRNA and the other containing Cre-ERT2-mCherry cassette as donor plasmid)	CRISPR/Cas9 (KI)	SD	Generation of six Cre driver rats allowing for the controlled gene expression or conditional KO in a temporal and spatial manner through the Cre-ERT2/*loxP* system.	*Wnt1* *Pomc* *Mnx1* *Drd1a* *G* *ad* *67* *Tie2*	Zhang et al. (2021) [150]
MI mRNA/gRNA/ssODN	CRISPR/Cas9 (KI)	SD F344	KI rats (carrying a T300A mutation) generated as a model for Crohn’s disease (CD); showing morphological abnormalities in both Paneth and goblet cells, but do not develop spontaneous intestinal permeability or inflammatory bowel disease; can be used for the study of both autophagy and CD susceptibility.	*Atg16l1*	Chesney et al. (2021) [151]
MI mRNA/gRNA	CRISPR/Cas9 (indels)	SD	KO rats generated to examine the function of dynein axonemal heavy chain 17 (*Dnah17*) in spermatogenesis; showing infertility with significantly decreased number of sperm; DNAH17 is critical for spermatogenesis.	*Dnah17*	Chen et al. (2021) [152]
MI PB vector carrying DdCBE pair/PB transposase mRNA	DdCBE (gene correction)	SD	DddA-derived cytosine base editor (DdCBE) was applied to explore the possible production of GE rats as a mitochondrial disease model with pathogenic mitochondrial DNA (mtDNA) mutations; MI of *piggy*Back transposon (PB) vector carrying DdCBE pair with PB transposase mRNA resulted in production of pups with mtDNA mutations with 36% efficiency.	*G7755*	Qi et al. (2021) [153]

## Data Availability

Not applicable.

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
