# Peer review of "Recent Advances in the Production of Genome-Edited Rats"

_ijms, 2022, doi:10.3390/ijms23052548_

Round 1
Reviewer 1 Report
The review is well written, it covers all major aspects of GE in rat and provides an updated reference point for the state of the art. Many acronyms are used - that means that not skilled readers could difficultly follow all text as they might go back to see the correct meaning of the used acronyms. It would be useful to simplify, for example, the figures by writing the complete words at least in the title of the different subsets in addition with the acronym.
Conclusions should be simplified. I would suggest to divide into "Perspectives" to indicate the future directions and the final conclusions where no references are usually included. This part should be substantially improved and rewritten.
Author Response
Reviewer-1:
The review is well written, it covers all major aspects of GE in rat and provides an updated reference point for the state of the art. Many acronyms are used - that means that not skilled readers could difficultly follow all text as they might go back to see the correct meaning of the used acronyms. It would be useful to simplify, for example, the figures by writing the complete words at least in the title of the different subsets in addition with the acronym.
Answer: Thank you for your helpful suggestion. As suggested, there are so many abbreviations in this text, which may confuse the readers for their well understanding. To avoid such misleading or make the readers understand the text in more comprehensive manner, we put the formal term before the acronyms (or abbreviation) as possible, which will be particularly evident in the revised Table 1. Furthermore, the abbreviations shown in the bottom of Table 1 of the previous text were moved into the “Abbreviation” section, just in front of “Reference” section in the revised text.
Conclusions should be simplified. I would suggest to divide into "Perspectives" to indicate the future directions and the final conclusions where no references are usually included. This part should be substantially improved and rewritten.
Answer: As suggested, the phrases shown in “Conclusion” section were divided into two groups, “6. Perspective” and “7. Conclusion” (please see L973 and L994 in the revised text). In Conclusion section, we removed references as possible.

Reviewer 2 Report
Sato and colleagues review recent advances in the production of genome-edited rats. They cover all established methods in great detail. The review is very well written. I have only a few suggestions for improvements.
As the authors were pioneering the i-GONAD technique, a table or paragraph providing a detailed protocol would be of enormous benefit for the community.
Some minor points are listed below:
Line 20: “which are important for biomedical research (such as generation of new human disease rat models and understanding the role of genes” is a repetition from above, please re-write
Line 56: There are some examples for GM rats already 20 years ago, e.g., PMID: 12206992, 9797336.
Table 1 Ref 63: Why the description is partly underlined?
Author Response
Reviewer-2:
Sato and colleagues review recent advances in the production of genome-edited rats. They cover all established methods in great detail. The review is very well written. I have only a few suggestions for improvements. As the authors were pioneering the i-GONAD technique, a table or paragraph providing a detailed protocol would be of enormous benefit for the community.
Answer: Thank you for your helpful suggestion. As for the technical aspect of GONAD/i-GONAD, the following phrases (Technically, this was simply achieved by inserting a glass micropipette through the oviductal wall under observation using a dissecting microscope, as shown in Figure 1C schematically. Next, a small amount of a solution (1-1.5 µL) containing GE components was injected into the oviductal lumen with the aid of a mouthpiece-controlled micropipette. Subsequently, in vivo EP was performed using tweezer-type electrodes and a square-wave pulse generator (T820; BTX Genetronics Inc.) under the EP condition of eight square-wave pulses with a pulse duration of 5 ms and an electric field intensity of 50 V.) are already shown in the section “3.3. GONAD-Based Production of GE Rats” in the revised text. This is a simple description, but appears to be suitable for grasping the whole image of the GONAD technique. As for the detailed protocols for GONAD/i-GONAD, we have provided the procedures in several papers (Gurumurthy et al., Nat. Protoc. 2019, 14, 2452–2482, doi:10.1038/s41596-019-0187-x; Ohtsuka and Sato, Dev. Growth Differ. 2019, 61, 306–315, doi:10.1111/dgd.12620), although all were involved in the production of genome-edited mice (but not rats). In a case that peoples want to get information of GONAD/i-GONAD, it would be convenient to see the above papers. Notably, the techniques shown in GONAD/i-GONAD (used for mice) can also be applied to rat genome manipulation, although the protocols for GONAD/i-GONAD specified for rats are now only available in the paper of Namba et al. [Dev. Growth Differ. 2021, 63, 439–447. doi: 10.1111/dgd.12746]. These are mentioned in the revised text (see L399 to L402).
Some minor points are listed below:
Line 20: “which are important for biomedical research (such as generation of new human disease rat models and understanding the role of genes” is a repetition from above, please re-write.
Answer: As suggested, the following phrase, “which are important for biomedical research (such as generation of new human disease rat models and understanding the role of genes)”, was deleted (please see L20 in the revised text).
Line 56: There are some examples for GM rats already 20 years ago, e.g., PMID: 12206992, 9797336. Unfortunately, production of genetically modified (GM) rats, as exemplified by knockout (KO) or knock-in (KI) rats, has not been possible until about 10 years ago.
Answer: As suggested, the phrases shown in L54 in the previous paper were rewritten, as follows: There are some examples for production of genetically modified (GM) rats reported approximately 20 years ago. These GM rats were transgenic (Tg) rats as “gain of function” model to examine the role of plasma membrane calmodulin–dependent calcium ATPase isoform 4 (PMCA4) cDNA under control of the cardiac-specific promoter [3] or sarcoplasmic reticulum Ca2+-ATPase (SERCA2a) under control of the ubiquitous promoter [4]. Unfortunately, production of GM rats, as exemplified by knockout (KO) or knock-in (KI) rats, has not been possible until about 10 years ago.
Table 1 Ref 63: Why the description is partly underlined?
Answer: This was only due to our simple mistake. This portion was corrected (see ref. 63 in the revised text).
